# Cdt1 stabilizes an open MCM ring for helicase loading

Jordi Frigola[1,*,†], Jun He[1,2,*,†], Kerstin Kinkelin[1], Valerie E. Pye[2], Ludovic Renault[3,†], Max E. Douglas[1], Dirk Remus[4], Peter Cherepanov[2], Alessandro Costa[3] & John F.X. Diffley[1]

ORC, Cdc6 and Cdt1 act together to load hexameric MCM, the motor of the eukaryotic replicative helicase, into double hexamers at replication origins. Here we show that Cdt1 interacts with MCM subunits Mcm2, 4 and 6, which both destabilizes the Mcm2–5 interface and inhibits MCM ATPase activity. Using X-ray crystallography, we show that Cdt1 contains two winged-helix domains in the C-terminal half of the protein and a catalytically inactive dioxygenase-related N-terminal domain, which is important for MCM loading, but not for subsequent replication. We used these structures together with single-particle electron microscopy to generate three-dimensional models of MCM complexes. These show that Cdt1 stabilizes MCM in a left-handed spiral open at the Mcm2–5 gate. We propose that Cdt1 acts as a brace, holding MCM open for DNA entry and bound to ATP until ORC–Cdc6 triggers ATP hydrolysis by MCM, promoting both Cdt1 ejection and MCM ring closure.

[1] Chromosome Replication Laboratory, The Francis Crick Institute, 1 Midland Road, London NW1 1AT, UK. [2] Chromatin Structure and Mobile DNA Laboratory, The Francis Crick Institute, 1 Midland Road, London NW1 1AT, UK. [3] Macromolecular Machines Laboratory, The Francis Crick Institute, 1 Midland Road, London NW1 1AT, UK. [4] Molecular Biology Program, Memorial Sloan Kettering Cancer Center, 1275 York Avenue, New York, New York 10065, USA. * These authors contributed equally to this work. † Present addresses: Institut d'Investigació Biomèdica de Girona Dr. Josep Trueta (IDIBGI), Parc Hospitalari Martí i Julià, 17190 Salt, Catalunya, Spain (J.F.); UCB Celltech, 216 Bath Road, Slough SL1 3WE, UK (J.H.); NeCEN, Gorlaeus Laboratory, Einsteinweg 55, Leiden 2333, The Netherlands (L.R.). Correspondence and requests for materials should be addressed to P.C. (email: Peter.Cherepanov@crick.ac.uk) or to A.C. (email: Alessandro.Costa@crick.ac.uk) or to J.F.X.D. (email: John.Diffley@crick.ac.uk).

The process of eukaryotic DNA replication begins during G1 phase with the loading of the minichromosome maintenance (MCM) heterohexamer comprising Mcm2–7 subunits into head-to-head double hexamers with double-stranded DNA passing through the length of a long central channel[1–3]. This topological loading requires the Origin Recognition Complex (ORC), Cdc6 and Cdt1 proteins. During S phase, the inactive double hexamer is converted by a set of firing factors into two active CMG (Cdc45-MCM-GINS) replicative helicases in which MCM acts as the ATP-dependent motor for DNA unwinding[4].

ATP binding and hydrolysis play distinct and crucial roles in MCM loading and ORC, Cdc6 and MCM all contain AAA+ ATPase family members[5]. ATP binding by ORC is required for its stable binding to origin DNA, ATP binding by Cdc6 is required for the formation of a stable ORC–Cdc6 complex at origins and ATP binding by MCM subunits stabilizes the MCM heterohexamer[3,6–8]. ATP hydrolysis by ORC and Cdc6 is not required for MCM loading[6,7] but ATP hydrolysis by Cdc6 plays an essential role in disassembling incomplete intermediate complexes[6,9]. MCM loading requires ATP hydrolysis by MCM subunits: mutation of arginine finger residues in several MCM subunits including Mcm2, 3, 5, 6 and 7 all inhibit loading to different extents[6,7]. Thus, it is important that MCM is bound to ATP before loading.

Previous work has shown that DNA enters the central channel of the MCM hexamer via the Mcm2–5 interface or 'gate'[10,11]. How the opening and closing of this gate is regulated to ensure DNA enters only at the correct time is unknown. In budding yeast, Cdt1 forms a stable complex with MCM before loading[12,13]. Cdt1 is not required for recruitment of MCM to ORC–Cdc6, but does contribute to the stability of MCM subunits in this recruited complex[9]. Cdt1 is released during loading, before double hexamer formation[14], and its release requires ATP hydrolysis by MCM subunits[6,7]. Here we use biochemistry and structural approaches to understand how Cdt1 contributes to MCM loading.

## Results

### Biochemical characterization of Cdt1–MCM interactions.
To understand how Cdt1 interacts with MCM, we first set out to determine which MCM subunits interact directly with Cdt1 using individual subunits expressed in *E. coli*. Figure 1a (arrowheads) shows that both Mcm6, as previously shown[15], and Mcm2 were co-immunoprecipitated with Cdt1 using an anti-Cdt1 antibody in a Cdt1-dependent manner. We next used glycerol gradients to examine complex formation between Cdt1, Mcm2 and Mcm6 (Supplementary Fig. 1a). Cdt1 formed stable binary complexes with Mcm2 and Mcm6 individually, and a ternary complex with Mcm2 and Mcm6, especially evident from the shift in Cdt1 sedimentation position. We used protein crosslinking with the bifunctional, amino reactive crosslinker bis(sulfosuccinimidyl)suberate (BS3), to determine which subunits Cdt1 interacts with in the intact MCM complex. After crosslinking Cdt1–MCM with limiting amounts of BS3, the complex was denatured with SDS and specific subunits were immunoprecipitated under denaturing conditions. Thus, only covalently attached proteins should co-immunoprecipitate under these conditions. We first used complexes in which various MCM subunits were tagged with GFP-FLAG (Supplementary Fig. 1b(i)). Crosslinked proteins were immunoprecipitated with anti-FLAG antibody and the presence of Cdt1 in the crosslinked products was examined using anti-Cdt1 antibody (Supplementary Fig. 1b(ii)). Supplementary Fig. 1b(ii) shows that Cdt1 crosslinked strongly to Mcm6 and less strongly to Mcm2 and Mcm4; Cdt1 did not crosslink to Mcm3 or Mcm7 under these conditions. Mcm5 was not

examined, because it is unstable when tagged at the carboxy terminus. We next used Cdt1–MCM in which Cdt1 was FLAG-tagged. After denaturing immunoprecipitation with FLAG antibody, four bands were seen above the 180 kDa marker by silver staining, which were not present in the absence of crosslinker, but appeared at low crosslinker concentration (Fig. 1b). We identified the proteins present in these bands by immunoblotting and mass spectrometry. Immunoblotting with Mcm2 or Mcm6 antibodies identified crosslinked complexes containing these proteins (Fig. 1b and Supplementary Fig. 1c), which was confirmed by mass spectrometry (Fig. 1b and Supplementary Table 1). The fastest migrating crosslinked band contained neither Mcm2 nor Mcm6, but was enriched for peptides from Mcm4 by mass spectrometry (Fig. 1b and Supplementary Table 1). From these experiments we conclude that Cdt1 interacts with Mcm6, Mcm2 and, to a weaker extent, Mcm4.

To determine how Cdt1 interacts with Mcm6, its strongest binding partner, we expressed MBP fusions containing different regions of Mcm6 (Fig. 1c(i)) and examined interaction with Cdt1 by amylose–agarose pull down. Figure 1c(ii) shows that Cdt1 did not interact with the amino terminus of Mcm6, but interacted strongly with a fragment containing the entire C-terminal half of Mcm6 and a shorter fragment lacking the AAA+ domain, containing only the C-terminal 228 amino acids of Mcm6. A protein containing all of Mcm6, except for this C-terminal domain, however, was still able to interact with Cdt1 (Supplementary Fig. 1d(i)), although not as well as the full-length protein (Supplementary Fig. 1d(ii)), indicating that Cdt1 interacts with Mcm6 at more than one position, including one in the C terminus and one within or near the AAA+ domain. This Mcm6ΔC protein could still form a ternary complex with Mcm2 and Cdt1 (Supplementary Fig. 1e(i)), and a high molecular-weight complex with the full complement of MCM subunits and Cdt1 (Supplementary Fig. 1e(ii)). Cdt1 inhibits the ATPase of the full MCM complex[16] (Supplementary Fig. 1f(i)), as well as that of MCM containing Mcm6ΔC (Supplementary Fig. 1f(ii)). Three pairs of MCM subunits, Mcm2–6, Mcm4–7 and Mcm7-3, exhibit ATPase activity on their own[17]. Figure 1d shows that, in addition to the intact MCM complex, Cdt1 also inhibits the ATPase of Mcm2–6 and Mcm4–7 but not Mcm7-3.

We next generated fragments containing the N-terminal (N), middle (M) and C-terminal (C) domains of Cdt1 in various combinations (Fig. 1e(i)), and used these fragments in pull-down experiments to determine which parts of Cdt1 are involved in MCM binding. As shown in Fig. 1e(ii), a fragment containing the 438 N-terminal amino acids of Cdt1 (Cdt1-NM) interacted with neither Mcm2 nor Mcm6, whereas a fragment containing the C-terminal 333 amino acids (Cdt1-MC) interacted with both Mcm2 and 6. Further division of the C terminus of Cdt1 showed that fragments containing amino acid residues 271–496 and 495–604 interacted with both Mcm2 and 6, suggesting there are multiple interactions between Cdt1 and MCM subunits.

We noticed during the course of analysing Cdt1 complexes containing different combinations of MCM subunits by gel filtration that Cdt1 appeared to influence interactions between Mcm2 and 5. An example of this is shown in Supplementary Fig. 1g. A stable tetrameric complex of Mcm6–2–5–3 (the order of contiguous subunits in the MCM hexamer) could be identified by gel filtration, peaking in fraction 5 (Supplementary Fig. 1g(i)); however, upon addition of Cdt1, this dissociated into two complexes: one containing Mcm2, 6 and Cdt1, peaking in fractions 3 and 4, and one containing Mcm3 and 5, peaking in fractions 6–7 (Supplementary Fig. 1g(ii)). To investigate this further, we incubated the tetrameric Mcm6–2–5–3 complex with Cdt1 immobilized on agarose beads. Figure 1f shows that only

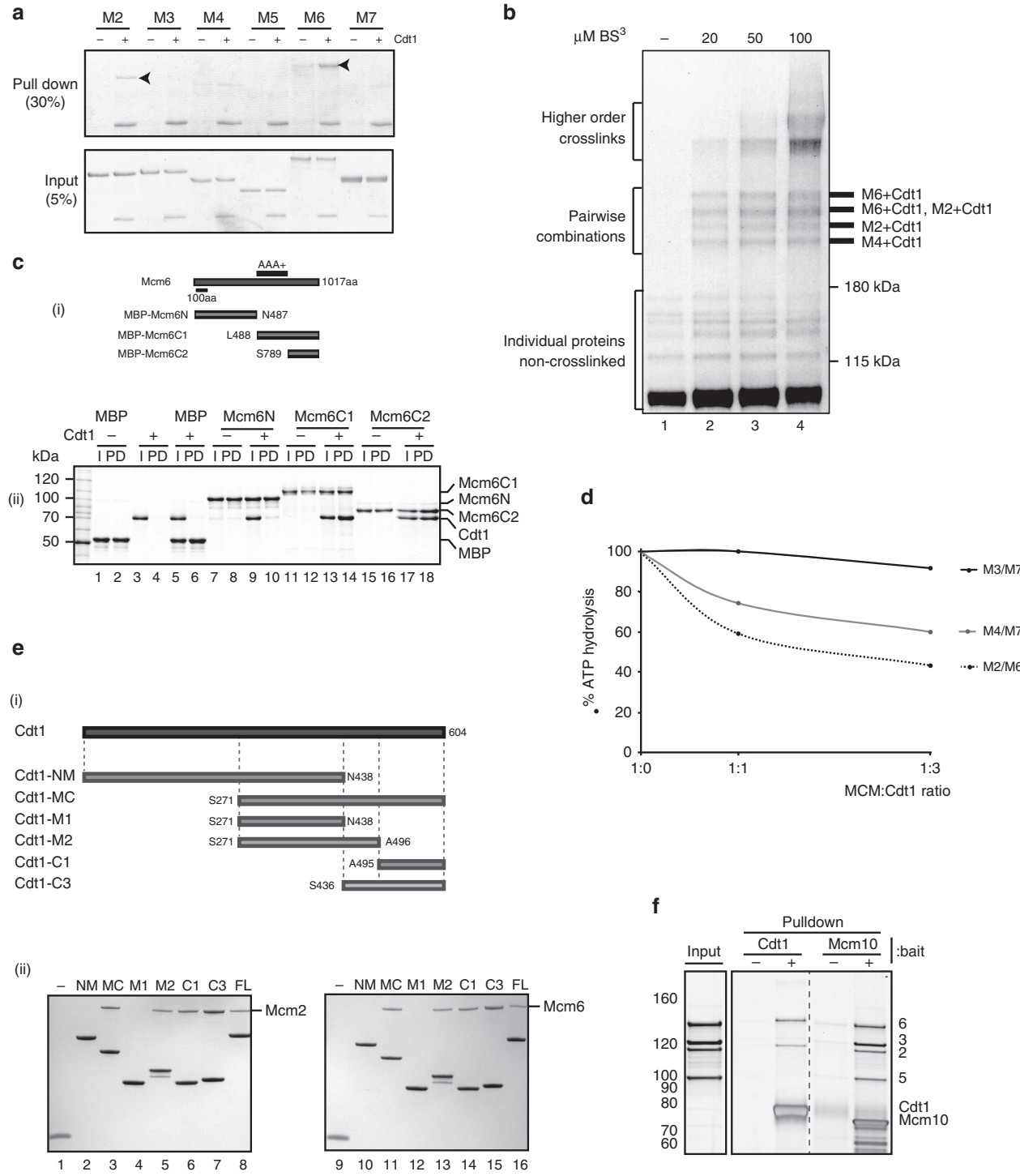

**Figure 1 | Biochemical characterization of Cdt1–MCM.** (**a**) Interaction of Cdt1 with individual MCM subunits after immunoprecipitation with anti-Cdt1 antibody (arrowheads highlight which individual Mcm interacts directly with Cdt1). (**b**) Crosslinking studies of MCM–Cdt1 FLAG complex. Denaturing immunoprecipitation of Cdt1-FLAG after crosslinking of this complex at indicated concentrations of crosslinker (BS3). Pairwise combinations were identified using immunoblotting and mass spectrophotometry, and summarized on the right. (**c**) Mapping Cdt1 interaction on Mcm6. (i) A series of Mcm6 fragments were fused to MBP tag. (ii) Amilose–agarose pull down of these fragments with the presence of Cdt1. Minus Cdt1 and MBP alone were used as a positive and negative control, respectively. Twenty per cent of the input and 50% of the pull downs were loaded on the gel. (**d**) Effect of Cdt1 on ATPase activity of MCM subunit pairs (M2/M6 = Mcm2-6; M4/M7 = Mcm4-7; M3/M7 = Mcm3-7). (**e**) Mapping domains of Cdt1 interacting with Mcm2 and Mcm6. (i) Cdt1 fragments fused to His-SUMO tag. (ii) His-tag pull downs with the presence of either Mcm2 (left) or 6 (right). His-SUMO protein was used as a negative control ( − ). (**f**) Pulldown of Mcm6-2-5-3 tetramer with immobilized Cdt1 or Mcm10.

Mcm2 and 6 were retained by Cdt1. We have previously shown that Mcm10, a protein required for helicase activation, also interacts with Mcm2 and 6, but not Mcm3 or 5, and that Cdt1 and Mcm10 compete with each other for MCM binding[18]. As shown in Fig. 1f, in contrast to Cdt1, Mcm10 pulls down all four subunits of the tetramer, despite not interacting directly with

Mcm3 or 5. Thus, Cdt1, but not Mcm10, destabilizes the Mcm2–5 interface after binding to Mcm2 and 6.

**X-ray crystallography of Cdt1.** To understand Cdt1–MCM at an atomic level, we determined crystal structures of two non-over-lapping fragments of *S. cerevisiae* Cdt1 spanning amino acid residues 1–438 and 495–604 (Supplementary Table 2 and Fig. 2). The structure of Cdt1(1–438) was refined using two independent crystal forms to resolutions of 2.7 and 2.1 Å. The fragment comprises a pair of well-defined domains corresponding to the N domain (residues 14–298) and the M domain (residues 299–430) (Fig. 2). The relative orientation between the domains is preserved in all four crystallographically independent copies of the protein chain observed in the crystal structures (Supplementary Fig. 2a), indicating a rigid linkage between the domains. The M domain belongs to the winged helix domain (WHD) family and is predictably similar to the geminin-binding M domain of mouse Cdt1 (Supplementary Fig. 2b). Perusal of the Dali structure comparison server[19] revealed a striking similarity between the Cdt1 N domain and Fe(II)/α-ketoglutarate-dependent dioxygenase superfamily of enzymes, such as AlkB and other DNA/RNA dealkylators[20], and the Jumonji C histone demethylase[21] (Supplementary Fig. 2c). However, the key His and Asp residues involved in chelation of the Fe(II) ion co-factor in the active sites of these enzymes are not conserved in the Cdt1 N domain (Supplementary Fig. 2c). Moreover, an α-helical hairpin (α6/α7), which extends the dioxygenase fold, occludes the degenerate active site of the N domain (Supplementary Fig. 2c). These observations indicate that the dioxygenase structural fold was repurposed for a non-catalytic function. The structure of Cdt1(495–604) revealed the C domain (Fig. 2), which is highly similar to the analogous WHD domain previously characterized in mouse Cdt1 (refs 22,23) (Supplementary Fig. 2b).

**The N terminus of Cdt1 contributes to MCM loading.** To examine the role of the dioxygenase-like N domain, we expressed and purified wild-type Cdt1, Cdt1MC (lacking the N domain), the hexameric MCM complex without Cdt1 and the heptameric, co-expressed Cdt1–MCM complex (Supplementary Fig. 3a). Cdt1–MCM complexes were reconstituted by incubation of MCM with an excess of Cdt1 or Cdt1MC before use. Figure 3a shows that MCM is recruited to DNA in the presence of ATP after low-salt wash similarly in the co-expressed Cdt1–MCM complex and in the Cdt1–MCM complex reconstituted from independently expressed MCM and Cdt1 (lanes 1 and 3). MCM loading onto DNA—defined by the generation of Mcm2–7 double hexamers bound to DNA after high-salt extraction of ORC and Cdc6 (refs 2,24,25)—occurred with both the co-expressed and reconstituted complexes, although it was routinely more efficient with the co-expressed complex (Lanes 2 and 4). The reconstituted complex of Cdt1–MCM recruited MCM, as well as

the complex containing wild-type Cdt1 (lane 5); however, loading was severely reduced with the truncated Cdt1MC (compare lanes 4 and 6). This is consistent with previous results from Takara and Bell[26], who showed reduced MCM loading by Cdt1 lacking its N terminus. Takara and Bell[26] further observed that MCM loaded by this truncated Cdt1 was less efficient in replication assays using S phase extracts than MCM loaded by full-length Cdt1 and this was confirmed by our experiments (Supplementary Fig. 3b). This defect could be due to the reduced MCM loading or could reflect some additional role for the N domain of Cdt1 in replication, which might be interesting, given its homology to dioxygenases. To address this we manipulated our loading reactions, so the truncated Cdt1 loaded the same amount of MCM as the full-length protein (Fig. 3b). Figure 3c shows that MCM loaded by the truncated Cdt1 is equally stable to high-salt extraction as the MCM loaded by wild-type Cdt1. If the N domain of Cdt1 had a role in replication beyond MCM loading, these MCM complexes should show reduced replication in extracts. However, as shown in Fig. 3d, Dbf4-dependent kinase (DDK)-dependent replication from these loaded MCM complexes was as efficient as from the complexes loaded with full-length Cdt1 (Fig. 3d, compare lanes 2 and 4). Therefore, the N domain of Cdt1 plays an important role in MCM loading and plays no role in replication downstream of MCM loading. The fact that this domain is essential *in vivo* may be due to this defect in loading.

**EM of Cdt1–MCM.** To understand better the function of Cdt1 in MCM loading, we used single-particle electron microscopy (EM) to characterize the MCM complex, either in isolation or bound to Cdt1 (Fig. 4). As judged from end-on view two-dimensional (2D) class averages of negatively stained particles, apo MCM appears to exist as a mixture of open and closed rings (Fig. 4a and Supplementary Fig. 4—as previously observed in ref. 27). When imaged in the presence of ATPγS, however, open MCM rings were not detected, suggesting that interaction with a slowly hydrolysable ATP analogue promotes ring closure (Fig. 4a and Supplementary Fig. 5, in agreement with refs 10,28). It is not formally possible, however, to distinguish closed rings from tilted views of open rings in 2D, especially when using low-resolution negative stain EM. To confirm that ATPγS-treated MCM particles were *bona fide* closed rings, we have reconstructed a three-dimensional (3D) structure, generated from all particles contributing to high-quality 2D averages, irrespective of their configuration. In these conditions, MCM appears to form a planar ring, with a topologically closed N-terminal tier and a slightly notched AAA+ tier (structure solved to 23.6 Å resolution, Fig. 4a and Supplementary Fig. 5). We then repeated the EM analysis using the Cdt1–MCM complex. In both the apo and the ATPγS-bound form, the heptameric complex appears to contain open MCM rings only, suggesting that Cdt1 has a role in stabilizing MCM in its open configuration (structures solved to 20.4 and 18.0 Å resolution, respectively, Fig. 4a and Supplementary Figs 6 and 7). Three-dimensional reconstruction of the Cdt1-bound MCM shows essentially the same structure in the apo and in the ATPγS-bound form, with a lock-washer ring configuration containing a discontinuity that spans both the N-terminal and the AAA+ tier of MCM (Fig. 4a and Supplementary Figs 6 and 7). An analogous topologically open form of MCM had been previously observed for the *Drosophila melanogaster*[27] and *Encephalitozoon cuniculi*[29] MCM, which were observed to form a left-handed spiral configuration. This was determined by tilt validation analysis and docking of the markedly chiral structure of the N-terminal MCM domain of an archaeal orthologue, available at the time[29]. In the *Drosophila*

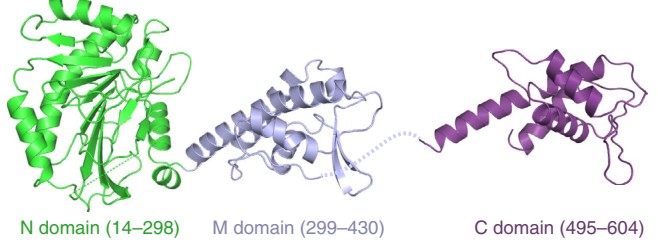

**Figure 2 | Crystal structures of the N-terminal and middle domains (left) and the C-terminal domain (right) portions of *S. cerevisiae* Cdt1.** The protein chains are shown as cartoons, with the N domain, M domain and C domain in green, blue and violet, respectively.

N domain (14–298)    M domain (299–430)    C domain (495–604)

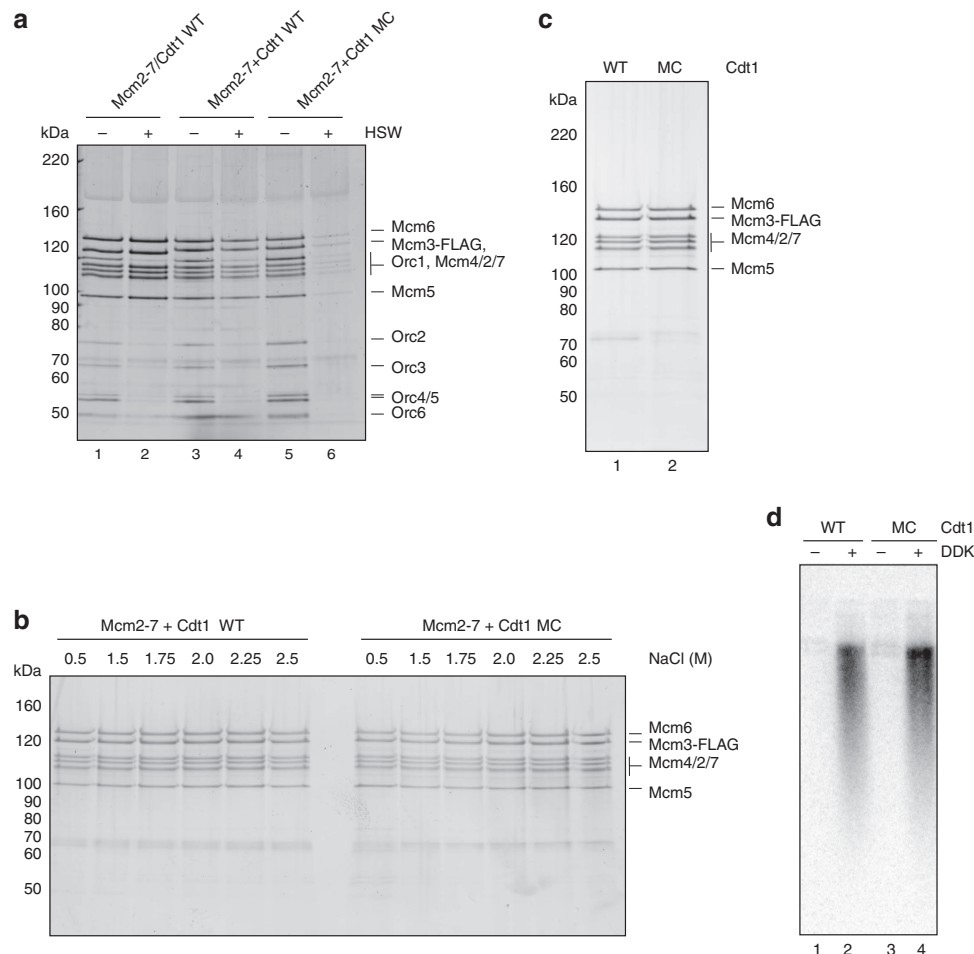

**Figure 3 | The N domain is important for MCM loading.** (**a**) Mcm2–7 recruitment ( − high-salt wash, HSW) and loading ( + HSW) on DNA analysed by SDS–PAGE and silver staining. Loading of co-expressed Cdt1–MCM complexes (lane 2) is more efficient than loading with reconstituted complexes of Mcm2–7 and Cdt1 WT (lane 4). Loading is dramatically reduced with Cdt1MC (lane 6). Recruitment of Mcm2–7 is not as dramatically affected (lanes 1, 3, 5). (**b**) Mcm2–7 loading on DNA analysed by SDS–PAGE and silver staining. Loaded Mcm2–7 is stable in high-salt washes up to 2.5 M NaCl, both when loaded with Cdt1 WT or Cdt1MC. (**c**) Mcm2–7 loading on DNA analysed by SDS–PAGE and silver staining (left), and (**d**) alkaline agarose gel (right) of replication products after in vitro replication assays using S-phase whole-cell extract. When adjusting levels of Mcm2–7 loading (left) for Cdt1 WT (lane 1) and Cdt1MC (lane 2)-loaded hexamers, DDK-dependent replication products and efficiency is comparable (right).

structure, N-terminal MBP tags were used to establish that the discontinuity in the MCM complex exists between Mcm5 and Mcm2 subunits[27,29]. Two atomic structures of yeast MCM have since become available (from the inactive double-hexameric MCM and the active Cdc45-MCM-GINS, CMG helicase) and we have used the latter in an atomic docking exercise (fitting the MCM N-terminal tier as a rigid body)[30,31]. Recognition of the characteristic propeller shape of the A domain of MCM immediately establishes the handedness of the Cdt1–MCM map (Supplementary Fig. 8 and Supplementary Movie 1), indicating that when Cdt1 associated MCM forms a left-handed spiral configuration similar to prior observations on the isolated MCM complex[29]. Docking of the MCM N-terminal domain (extracted from either the MCM double hexamer or the CMG helicase) into either the apo or the ATPγS-bound Cdt1–MCM EM map assigns the discontinuity at the Mcm2–5 interface, in agreement with previously published data[3,10,28,29], as well as with biochemical data presented here (Fig. 4b and Supplementary Fig. 8). To optimize the atomic docking and accurately model the MCM discontinuity, the N-terminal domain of MCM was separated into two distinct subcomplexes (Mcm5–3–7 and Mcm2–4–6), which were used for rigid body fitting (Fig. 4c and Supplementary Fig. 9). Superposition of full-length MCM subunits, followed by

local adjustment of the isolated AAA + /WH domains of Mcm5–3–7 and Mcm2–4–6 were used to complete the assignment of MCM domains and model the Mcm2–5 discontinuity in the AAA + tier (Fig. 4d and Supplementary Fig. 9). The marked spiral structure of the Cdt1-associated MCM causes a gross misconfiguration of the Mcm2–5 active site (Fig. 4d and Supplementary Fig. 9), whose reconstitution would require planarization and tightening of the MCM ring. The resulting molecular model for the open MCM agrees with recently published cryo-EM data[32]. Once MCM subunits were assigned in the Cdt1–MCM map, unoccupied density could be detected, intimately contacting the Mcm6 and Mcm2 subunits, and radially projecting from Mcm2 (Fig. 4b). We assign this density to Cdt1. By docking the crystal structure of NM Cdt1 into the Cdt1 density (Fig. 4b and Supplementary Fig. 9), an MCM interaction can be modelled, whereby the middle domain of Mcm2 is sandwiched between the A-domain and AAA + domain of Mcm2 and extends to contact the A-domain of Mcm6, following the outer perimeter of the MCM N-terminal domain (Fig. 4c-e). Conversely, the N domain of Cdt1 projects away from the MCM ring and does not appear to be involved in any contact with the helicase motor (Fig. 4c,d and Supplementary Movie 1). The domain interaction studies presented in Fig. 1e support this

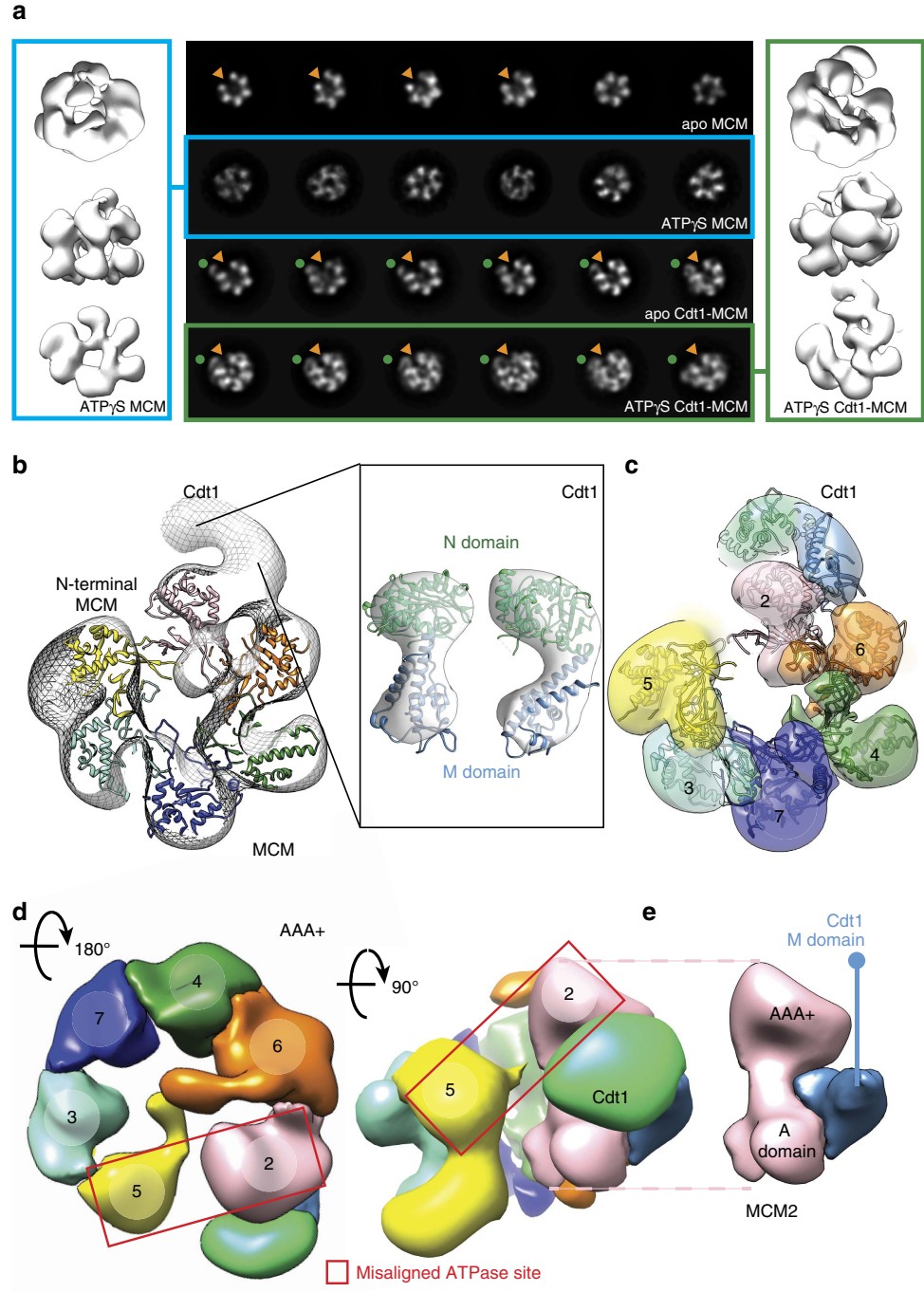

**Figure 4 | Negative-stain electron microscopy of yeast MCM in the absence and presence of Cdt1.** (**a**) Two-dimensional averages of free and Cdt1-associated MCM, in the apo and ATPγS-bound state (central panel). A red arrowhead points the MCM gate. ATPγS binding causes gate closure in the free MCM but not in the Ctd1-bound state. A red arrowhead points the MCM gate. Three-dimensional reconstruction of a topologically closed, ATPγS-bound MCM (left panel) and an open ATPγS-bound Cdt1–MCM (right panel). (**b**) Atomic docking of the N-terminal MCM is used to identify the register of MCM proteins. Unoccupied density on the side of the MCM ring is assigned to Cdt1. (**c**) The N-terminal domain of Cdt1 (green) is not involved in MCM interactions. The middle domain of Cdt1 contacts the A domains of Mcm2 and Mcm6. (**d**) Cdt1 binding stabilizes a left-handed spiral configuration of MCM. In this state, the Mcm2–5 ATPase interface is grossly misaligned. The middle domain of Cdt1 is sandwiched between the Mcm2 N domain and AAA+ domain. (**e**) Detail of the Mcm2—Cdt1 M domain interaction. The M domain of Cdt1 (light blue) is sandwiched in between the A and AAA+ domains of Mcm2 (pink).

assignment. Notably, the connectivity between N domain and M domain observed in our 2.1 Å crystal structure differs drastically from the atomic model based on a 7 Å resolution cryo-EM structure of MCM—Cdt1 (EMD-6671), in that N domain (which appears pseudo two-fold symmetric at 7 Å resolution) is ∼180° rotated with respect to M domain[32] (Supplementary Fig. 10). Although the C-terminal WHD of Cdt1 could be accommodated

in our EM structure, interacting with the WHD of Mcm6, we have not modelled this region using our negative stain data, due to the limited resolution. Our CTD-Cdt1 crystal structure can however be docked into the 7 Å resolution cryo-EM structure of MCM-Cdt1, indicating that the C-terminal WHD of Cdt1 interdigitates between the A domains of Mcm4 and Mcm6, in agreement with two recently published atomic models[32,33]. In

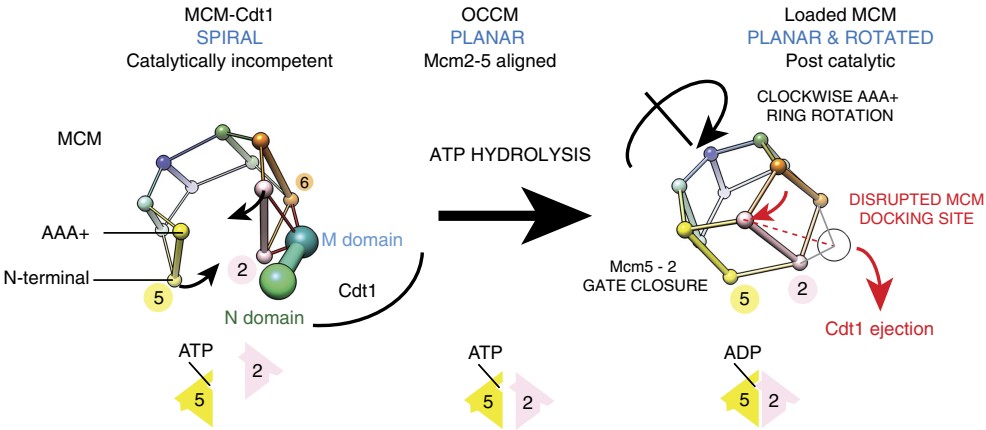

**Figure 5 | A mechanism for ATPase-mediated Cdt1 ejection during MCM loading.** Cdt1 binds to the side of Mcm2 and Mcm6, interacting both with N-terminal and AAA+ ATPase elements. Cdt1 stabilizes a topologically open left-handed spiral configuration of MCM, with a discontinuity between the Mcm5 and Mcm2 subunits, disrupting a functionally essential AAA+ ATPase site. DNA recruitment mediated by ORC and Cdc6 (OCCM formation) planarizes the MCM ring, reconstituting the Mcm5–2 ATPase site, which becomes competent for ATP hydrolysis. Conversion to a post-catalytic, loaded MCM involves the clockwise rotation of the AAA+ ATPase tier. This conformational change probably disrupts a Cdt1 docking site on Mcm2, promoting ATPase-dependent Cdt1 ejection.

summary, Cdt1 makes extensive contacts with the outer perimeter of MCM, notably being sandwiched between the A and AAA+ domains of Mcm2. Comparative analysis of the nucleotide-bound Cdt1–MCM form with the post-catalytic MCM double hexamer (corresponding to the loaded form[30]) highlights a gross reconfiguration of the AAA+ tier, which would occur upon transitioning from an open lock-washer to the loaded MCM configuration. This is best visualized using the centre of mass of A and AAA+ domains of the six Mcm subunits (Fig. 5). The ATPase tier globally rotates clockwise, whereas the N-terminal domain of Mcm5 moves counterclockwise to seal the Mcm2–5 discontinuity. In the loaded MCM form, the AAA+ domain of Mcm2 moves in close proximity to Mcm5, reconstituting a key active site that is essential for the ATPase function of MCM, and concomitantly disrupting the Mcm2–AAA+/Cdt1 interaction, one major Cdt1–MCM contact point (distance between centre of mass of Mcm2 AAA+ and Cdt1 middle domain is 49 Å in Cdt1–MCM and would be 61 Å in the loaded MCM form, Fig. 5)[30].

## Discussion

We propose that Cdt1 stabilizes a conformation of the MCM ring in which the Mcm2–5 gate is held open to allow duplex DNA passage required for helicase loading. In this state, the Mcm2–5 opening is much wider than in the ORC–Cdc6–Cdt1–MCM complex, suggesting that ORC plays a role in closing, not in opening the MCM structure. We further propose that, by holding MCM in this conformation, ATP hydrolysis is inhibited, which helps ensure MCM-Cdt1 is bound to ATP for subsequent loading (Fig. 5). Previous work in a variety of systems has led to the idea that loading of ring-shaped multi-subunit protein complexes around duplex DNA can occur via two mechanisms: ring making, in which individual protomers assemble around DNA to form a ring, and ring breaking, in which a pre-formed ring is transiently destabilized at one inter-protomer interface by a loading factor before loading[34]. Our results suggest that MCM loading has aspects of both ring making and ring breaking. In agreement with previous work on *Drosophila* MCM[27,29], we found that yeast MCM exists as a mixture of open and closed rings and the closed form is stabilized by ATPγS binding. Cdt1 stabilizes the open form of MCM, which exists as a left-handed spiral. Thus, in contrast to the DnaC loader of the bacterial replicative helicase,

DnaB[35], or the δ-subunit of the *Escherichia coli* clamp loader[36], which act as classic ring breakers, Cdt1 appears to selectively bind and stabilize the open MCM form, by associating with the immediate neighbour (Mcm2–6) of a preformed Mcm2–5 gate. Therefore, Cdt1 acts more like an open-ring stabilizer than a canonical ring breaker[34]. Moreover, once MCM is recruited to ORC–Cdc6 at origins, Cdt1 prevents dissociation of Mcm3,5,7 (ref. 9), suggesting an element of ring making or ring stabilizing in this process.

MCM must hydrolyse ATP during loading[6,7]. We propose that the inhibition of ATPase activity by Cdt1 plays an important role in keeping MCM in an appropriately ATP-bound form required for subsequent loading. Our Cdt1–MCM structure, in which MCM subunits are in a left-handed spiral and the Mcm2–5 active site is disrupted, helps explain how Cdt1 association inhibits the MCM ATPase activity. We note that multiple MCM assemblies have been characterized by crystallography and electron microscopy with left handed AAA+ spiral, which have been described as inactive forms of the ATPase[29,31,32,37].

Something must occur during loading which counteracts this inhibition of ATPase by Cdt1 and allows ATP hydrolysis by MCM. We propose that this involves the transition from the open spiral with misaligned Mcm2–5 ATPase elements to a planar structure where Mcm2 and Mcm5 are properly configured for hydrolysis. We suggest that it is the binding of Cdt1–MCM to ORC–Cdc6 that promotes this transition. This is supported by the cryo-EM structure of the ATPγS stabilized ORC–Cdc6–Cdt1–MCM loading intermediate in which the MCM subunits are in a planar configuration, containing a more aligned, yet still misconfigured, Mcm2–5 ATPase centre[33,38]. We suggest that ATP hydrolysis by the neighbouring MCM sites promotes the reconstitution of the bipartite Mcm2–5 ATPase site (that is, ring closure) and concomitant Cdt1 ejection, en route to MCM loading[6]. This is consistent with the finding that MCM subunit mutants defective in ATP hydrolysis do not eject Cdt1 efficiently[6,7] and is also consistent with recent single molecule experiments showing concomitant release of Cdt1 and MCM ring closure[39]. Multiple ATPase centres including 2–5, 5–3 and 3–7 are required for MCM loading[6,7], which is unexpected given that Cdt1 contacts only the Mcm2, 4 and 6 subunits. Comparative analysis of the pre-catalytic Cdt1–MCM complex and the post-

catalytic MCM double hexamer[30] suggests a mechanism for this process. Transition from an open (Cdt1–MCM) to a locked MCM ring (as observed in the MCM double hexamer) involves the clockwise rotation of the AAA+ ATPase tier with respect to the N-terminal domain of MCM and this rotation probably disrupts a Mcm2–Cdt1 interaction point, which appears to be the main Cdt1 docking site in our structure (Fig. 5 and Supplementary Movie 2). Combined, our biochemical and structural data explain how Cdt1 keeps the MCM ring open and the ATPase inactive, before ORC-directed DNA association, which is concomitant with a major reconfiguration of the ATPase tier of MCM and Cdt1 ejection. Previous work has shown that Mcm3 binding promotes ATP hydrolysis by ORC–Cdc6 (ref. 9); we propose there is a mutual activation of the MCM ATPase by ORC–Cdc6 binding as well.

The N domain of Cdt1 is a dioxygenase, which has been catalytically inactivated during evolution. In metazoa, this N domain is much shorter and shows little or no sequence similarity to the yeast protein, and thus may be structurally unrelated to the yeast N domain. The N domain of the yeast protein plays an important role in MCM loading, but makes little or no contact with MCM in our structure. It is possible that this domain transiently contacts ORC–Cdc6, DNA or MCM in the early stages of helicase loading.

## Methods

**Reagents.** All yeast strains are described in Supplementary Table 3, all plasmids in Supplementary Table 4 and all oligonucleotides in Supplementary Table 5.

**Glycerol gradients.** Fifteen micrograms of Mcm2, 6 and Cdt1 were loaded onto 5 ml centrifuge tubes (Beckman) containing 15–35% glycerol gradient of buffer A (25 mM Hepes, pH 7.6, 1 mM EDTA, 0.05% NP-40, 10% glycerol and 0.1 M potassium glutamate). Gradients were centrifuged at 42,000 r.p.m. for 16 h (SW55 Ti rotor (Beckman)). Twenty-seven aliquots of 175 μl were taken manually. Of these aliquots, 20 μl were loaded onto SDS–PAGE gel and visualized by silver staining.

**Pull downs.** Purified Mcm2–7 subunits (0.5 μM) were individually incubated at room temperature for 60 min with purified Cdt1 (0.5 μM) or a buffer control in 100 μl of buffer A/2 mM 2-mercaptoethanol. Reactions were subjected to immunoprecipitation for 2 h at 4 °C with ~5 μg of polyclonal αCdt1 antibody pre-bound to protein A sepharose. After two washes with 0.5 ml binding buffer each, beads were resuspended in SDS sample buffer and samples analsed by SDS–PAGE and Coomassie staining.

Mcm6 MBP fragments and Cdt1 proteins were dialysed against buffer A for 2 h at 4 °C. Each pull-down reaction contained 10 μg of each protein. Once mixed, proteins were incubated for 60 min at 4 °C in a rotating wheel. Ten microlitres of slurry amylose-agarose beads (NEB) prewashed with buffer A was used to pull down MBP-Mcm6 fragments after an incubation of 30 min at 4 °C. Minus Cdt1 and MBP were used as positive and negative controls, respectively. Beads were collected and washed twice with buffer A. Fifty per cent of the final volume was loaded onto SDS–PAGE gel and proteins were stained with Coomassie stain.

Cdt1-His-SUMO fragments together with Mcm2 and 6 were dialysed against buffer B (25 mM Hepes-KOH pH 7.6/0.02% NP-40/10% glycerol/0.1 M K glutamate) for 2 h at 4 °C. Ten micrograms of each dialysed protein was used per pull-down reaction. Once mixed, proteins were incubated for 60 min at 4 °C in a rotating wheel. Five microlitres of slurry Ni-NTA magnetic beads (Qiagen) prewashed with buffer B was used to pull down Cdt1-His-SUMO fragments, after an incubation of 30 min at 4 °C. His-SUMO tag alone was used as a negative control. Beads were collected and washed with buffer B twice. Fifty per cent of the final volume was loaded onto SDS–PAGE gel and proteins were stained by Coomassie stain.

Cdt1 FLAG together with Mcm6 fl and ΔC were dialysed against buffer B for 2 h at 4 °C. Ten micrograms of each dialysed protein was used per reaction. Once mixed, proteins were incubated for 60 min at 4 °C in a rotating wheel. Five microlitres of slurry M2 agarose FLAG beads (Sigma) prewashed with buffer B was used to pull down Cdt1 FLAG after an incubation of 30 min at 4 °C. Beads were collected and washed twice with buffer B. FLAG peptide (Sigma) was added to a final concentration of 0.5 M and the mixture was further incubated for 20 min at room temperature. Supernatant was collected and loaded onto SDS–PAGE gel and bound proteins were visualized by Coomassie stain.

Equimolar amounts of purified Mcm 6, 2, 5 and 3 were combined in 40 μl of buffer B/5 mM Mg acetate. After dialysis for 2 h, sample was run over a Superdex 200 size-exclusion column preequilibrated in buffer B/5 mM Mg acetate and

fractions containing tetrameric Mcm 6, 2, 5 and 3 complex pooled. The tetramer was incubated for 2 h on ice with Cdt1 or Mcm10 prebound to M2 anti-FLAG agarose (Sigma) and anti-T7-tag agarose (Abcam), respectively, and resins washed 2× with buffer B/5 mM Mg-acetate. Cdt1 complexes were eluted with the same buffer supplemented with 0.25 mg ml$^{-1}$ FLAG peptide, trichloroacetic acid (TCA) precipitated and resuspended in SDS-loading buffer, whereas Mcm10 complexes were released into SDS-loading buffer directly by boiling. Samples were separated on 3-8 % Tris-acetate gels (Biorad) and visualized by silver staining.

**Protein crosslinking.** Twenty micrograms of each MCM-GFP,FLAG–Cdt1 complexes were crosslinked with 25 μM BS3 (Thermo Fisher). However, with MCM–Cdt1 FLAG complex, 200 μg of protein were used for each BS3 concentration (20, 50 and 100 μM). After 30 min incubation at room temperature with buffer B, the reaction was stopped using buffer Tris-HCl pH 7.5 at a final concentration of 50 mM. The sample was denatured using 1% SDS, 50 mM Tris-HCl pH 8.0 and 1 mM fresh dithiothreitol (DTT) solution and 5 min incubation at 95 °C. The sample was diluted with RIPA buffer (50 mM Tris-HCl pH 8.0, 0.5% Deoxycholate, 1% NP40, 300 mM NaCl and 1 mM fresh DTT) and incubated with 600 μl slurry of M2 agarose flag beads (Sigma) for 1 h at room temperature. Beads were collected and washed twice with 0.1 and 0.5 M NaCl, respectively. 3XFLAG peptide (Sigma) was added to a final concentration of 0.5 M and incubated for 20 min at room temperature. Supernatant was collected and submitted to TCA precipitation. Crosslinked products were separated using 3–8% Tris acetate gels (Biorad), after running for 90 min at 180 V. Crosslinked products of MCM-GFP,FLAG–Cdt1 complexes were characterized by western blotting against Cdt1 and FLAG (input control), whereas crosslinked pairwise combinations of MCM–Cdt1 FLAG complex were further identified using both immunoblotting and mass spectrophotometry.

**ATPase assays.** Ten micrograms of each Mcm subunit was used to form the following dimers: 3/7, 4/7 and 2/6. These pairwise combinations were dialysed against buffer A for 2 h at 4 °C and they were fractionated over a Superdex 200 PC 3.2/30 column (GE Healthcare), pre-equilibrated in buffer A. Fractions containing the different dimers were pooled and used for ATPase assays. Five picomoles of each dimer was mixed with different molar ratios of Cdt1, as indicated in Fig. 1d. Reactions were carried out in a buffer containing the following: 25 mM Hepes-KOH, pH 7.6/0.1% NP-40/5 mM Mg(OAc)$_2$/1 mM EDTA/1 mM EGTA/ 100 mM K-acetate/5% Glycerol/1 mM DTT/100 μM ATP (including 2.5 μCi of [α-$^{32}$P]ATP). After 20 min at 30 °C, reactions were stopped by spotting 1 μl of each reaction on PEI cellulose TLC plates (CamLab). The cellulose membrane was developed in 0.6 M Na$_2$HPO$_4$/NaH$_2$PO$_4$ pH 3.5 and quantified on a Phosphor-imager (GE Healthcare).

The MCM full complex with increasing amounts of Cdt1 was obtained mixing MCM complex (from yJF39) with recombinant Cdt1. Typically, 10 pmol of this mixture was used per assay. MCM complexes with either Mcm6 fl or ΔC were obtained mixing recombinant individual Mcm's. Briefly, Mcm's were predialysed in buffer A and fractionated over Superdex 200 PC 3.2/30 column (GE Healthcare), pre-equilibrated in buffer A. Fractions containing equimolar amounts of individual Mcm's were pooled, quantified and used for the ATPase assays, together with recombinant Cdt1 at 3:1 molar ratio (Cdt1:MCM).

**MCM loading.** MCM recruitment and loading was carried out essentially as described[2,9]. ARS1-containing plasmids (pBluescript KS (+) ARS1 WT) were randomly biotinylated using the PHOTOPROBE (Long Arm) Biotin Kit (Vector Laboratories) according to manufacturer's instructions and immobilized on streptavidin M-280 magnetic DNA beads. As biotinylation efficiency was estimated only to be ~10–20%, 1 μg of DNA was coupled per 5 μl bead slurry. Immobilized DNA was incubated with Mcm2–7–Cdt1 complexes, ORC and Cdc6 in the presence of ATP at 37 °C for 20 min. DNA beads were then washed either twice with low salt wash buffer (25 mM Hepes-KOH pH 7.6, 300 mM K-acetate, 0.02% NP-40, 5 mM Mg(OAc)$_2$, 10% glycerol) when analysing recruitment, or once with high-salt wash buffer (25 mM Hepes-KOH pH 7.6, 500 mM NaCl, 0.02% NP-40, 5 mM Mg(OAc)$_2$, 10% glycerol) and once with low-salt wash buffer when analysing loading. DNA beads were then treated with MNase at 30 °C for 10 min and the supernatant was analysed by SDS–PAGE. For reconstituted Cdt1–MCM complexes, Mcm2–7 was incubated with a fivefold molar excess of Cdt1 WT or Cdt1 ΔN for 10 min on ice in the presence of 10 mM ATP before loading.

For achieving similar levels of loading with Cdt1MC, protein amounts were adjusted as follows (amounts for loading with Cdt1 WT in brackets): 13.5 (4.5) pmol Mcm2–7, 67.5 (22.5) pmol Cdt1MC and 1.5 (0.5) pmol Orc. Loading reactions were carried out at 30 °C for 60 min (20 min when using Cdt1 WT).

**DNA replication assays with S-phase whole-cell extract.** S-phase whole-cell extract was prepared and in vitro DNA replication assays were carried out essentially as described[40]. MCM was loaded onto ARS1-containing plasmids as described above. Next, supernatant of the reaction was removed and the beads were resuspended in a reaction mix containing 5× bead buffer (125 mM Hepes-KOH pH 7.6, 50 mM Mg(OAc)$_2$, 0.1% NP-40, 212.5 mM K-glutamate, 25% glycerol), 1 mM DTT, 5 mM ATP, 1 mM spermine (Sigma-Aldrich) and distilled water. The

reaction was supplemented with purified DDK and incubated at 25 °C with agitation for 15 min. The supernatant of the reaction was removed and beads were resuspended in a reaction mix containing 20× replication buffer (800 mM Hepes-KOH pH 7.6, 160 mM MgCl₂), 1 mM DTT, 5 mM ATP, 100 μM dATP/dCTP/dTTP/dGTP (Invitrogen) and 200 μM CTP/GTP/UTP (Invitrogen), 40 mM creatine phosphate (Calbiochem) and 10 μg creatine phosphokinase (Calbiochem). yKO3 S-phase extract (750 μg) was added last. The reaction was then incubated at 25 °C with agitation for 20 min.

Replication reactions were quenched by addition of 20 mM EDTA and beads were washed with TE buffer, before resuspending in 5 mM EDTA and then adding NaOH and sucrose to 50 mM and 1% w/v, respectively. Samples were incubated at room temperature for 30 min before products were separated through 1% alkaline agarose gels in 30 mM NaOH, 2 mM EDTA for 16 h at 25 V. Gels were fixed with 5% cold TCA, dried onto chromatography paper (Whatman) and autoradiographed with Amersham Hyperfilm-MP (GE Healthcare).

**X-ray crystallography.** *Saccharomyces cerevisiae* Cdt1 fragments spanning residues 1–438 (N domain and M domain) or 495–604 (CTD) were expressed in *E. coli* as fusions to hexahistidine-SUMO either in native or selenomethionine (SeMet) derivative form. The recombinant proteins were isolated from bacterial extracts by batch absorption to NiNTA agarose (Qiagen). Following digestion with SUMO protease and ion exchange chromatography, untagged proteins were polished by size-exclusion chromatography through a Superdex 200 16/600 column equilibrated in 200 mM NaCl, 25 mM Tris-HCl pH 7.4. Crystals were grown at 18 °C in hanging drops made by mixing 1 μl protein (10 mg ml⁻¹) and 1 μl reservoir solution.

Cdt1(1–438; NM) was crystallized in two forms. Crystals of form 1 were obtained using SeMet containing protein and reservoir solution of 30% glycerol (v/v), 20% polyethylene glycol-4000 (w/v), 20% isopropanol (v/v), 0.1 M Tris-HCl pH 8.5. X-ray diffraction data were collected at 100 K at a wavelength of 0.97934 Å on the beamline I04 of the Diamond Light Source (Oxfordshire, UK) and processed using XDS[41] and Aimless[42] via the Xia2 automatic pipeline[43]. The phases were determined by single-wavelength anomalous dispersion using phenix.autosol software[44], which identified 22 Se sites resulting in a figure of merit of 0.281 and overall score of 39.6 ± 11.31 in space group $P2_12_12_1$; 667 residues were automatically built giving a model-to-map correlation coefficient of 0.49, $R_{free}$/$R_{work}$ of 0.491/0.449 and a readily interpretable electron density map. The model was improved by auto-rebuilding using phenix.autobuild[44], resulting in a model containing 988 residues belonging to 3 copies of Cdt1(1–438) in asymmetric unit. Final rounds of manual building were performed in Coot[45] and refined using phenix.refine[44]. Crystal form 2 of Cdt1(1–438) was obtained using native protein and reservoir solution containing 10.5% polyethylene glycol-8000 (w/v), 18% glycerol (v/v) and 0.5 M lithium sulphate. Glycerol concentration was increased to 30% (v/v) before cryo-cooling in liquid nitrogen. X-ray diffraction data were collected on BM14 at the European synchrotron radiation facility (ESRF, Grenoble, France) and processed using XDS and Aimless via Xia2. One polypeptide from crystal form 1 was used as a search model for molecular replacement in Phaser[46] and the structure was refined using Coot[45] and phenix.refine.

Ctd1(495–604) spanning the CTD was produced in SeMet form and crystallized in 1.8 M ammonium sulphate, 20% glycerol (v/v), 0.1 M MES-NaOH, pH 6.5. X-ray diffraction data were acquired on ID23-1 beamline of the ESRF at 100 K and a wavelength of 0.97246 Å. The data were processed using XDS and Aimless via Xia2. The structure was solved using phenix.autosol, which identified 17 Se sites resulting in a map with a figure of merit of 0.34; 192 amino acid residues were auto-built giving a model-to-map correlation coefficient of 0.82 and a readily interpretable map in space group $P2_12_12_1$. Manual rebuilding in Coot and refinement in phenix.refine resulted in the final structure, which contains two molecules per asymmetric unit. Geometry of the final structures was assessed using MolProbity. Data collection and structure refinement statistics are given in Supplementary Table 2, and representative electron density maps for each domain are shown in Supplementary Fig 2d–f.

**Electron microscopy.** Carbon-coated grids for negative stain EM were prepared as follows. Carbon was evaporated onto freshly cleaved mica using a Q150TE coater (Quorum Technologies) and floated onto 400-mesh copper grids (Agar Scientific). Dried carbon grids were glow discharged for 30 s at 45 mA using a 100× glow discharger (Electron Microscopy Sciences). Four microlitres of sample were applied onto the grid and incubated for 30 s, before being partially blotted. The grid was sequentially laid on top of two drops (75 μl) of 2% uranyl formate solution (w/v), incubated for 30 s and then blotted dry. Grids were stored at room temperature before imaging. Micrographs of negatively stained preparations were collected on a JEOL-2100 electron microscope (JEOL) operated at 120 kV. Images were recorded at a nominal magnification of ×42,000 on an Ultrascan 4 k × 4 k charge-coupled device camera (Gatan), resulting in a 2.7 Å pixel size on the specimen level. Micrographs were collected with a 1–2 μm defocus range and an electron dose of 20 e⁻/Å². One hundred and thirty-one micrographs were collected for the apo MCM sample, 191 for ATPγS-MCM, 155 for apo Cdt1–MCM and 608 for ATPγS–Cdt1–MCM. Contrast transfer function estimation was performed with CTFFIND3 (ref. 47) and micrographs were phase-flipped with Bsoft[48]. Particles were selected semi-automatically using the e2boxer programme from the EMAN2 package[49].

Particles (33,862) were collected for the apo MCM sample, 38,924 for ATPγS-MCM, 41,013 for apo Cdt1–MCM and 123,753 for ATPγS-Cdt1–MCM. Individual particles were extracted and subjected to reference-free 2D average using Relion 1.3 package[50]. To reconstruct a 3D structure, an initial 3D model of the hexameric Mcm2–7 closed ring was created by segmenting one Mcm2–7 ring from the loaded double hexamer structure[2]. A 60 Å low-pass filter was applied onto this starting model to minimize bias during subsequent 3D image processing (Supplementary Fig. 11). This feature-less volume was used for 3D classification and 3D auto-refinement using RELION 1.3 package[50]. Automatic atomic docking, map segmentation (using the Color zone and Segment option) and 3D-EM figures were performed with UCSF Chimera[51].

**Data availability.** Crystal structures and structure factors have been deposited with the PDB under accession codes 5ME9, 5MEA and 5MEB. A construct spanning isolated Cdt1 middle domain was also crystallized and the resulting structure is deposited under accession code 5MEC. EM maps were deposited with the EMDB under accession number 3681 (ATPγS MCM), 3679 (apo MCM-Cdt1) and 3680 (ATPγS MCM-Cdt1).

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

## Acknowledgements

We thank the staff of the Diamond Light Source I04, and ESRF BM14 and ID23-1 beamlines for assistance with X-ray data collection. We are grateful to the Proteomics Laboratory for mass spectrometry. This work was supported by the Francis Crick Institute, which receives its core funding from Cancer Research UK (FC001061, FC001065 and FC001066), the UK Medical Research Council (FC001061, FC001065 and FC001066) and the Wellcome Trust (FC001061, FC001065 and FC001066). This work was also funded by a Wellcome Trust Senior Investigator Award (106252/Z/14/Z) and a European Research Council Advanced Grant (669424-CHROMOREP) to J.F.X.D. and an EMBO Long Term Fellowship (European Commission FP7, Marie Curie Actions, EMBOCOFUND2012, GA-2012-600394) to K.K.

## Author contributions

J.F., K.K., M.E.D., D.R. and J.F.X.D. designed the biochemical experiments in Figs 1 and 3, and Supplementary Information, and J.F., K.K., D.R. and M.E.D. performed these experiments. J.F., V.E.P. and P.C. designed and performed crystallization studies, and J.H., L.R. and A.C. designed and performed E.M. experiments. J.F.X.D. coordinated writing of the manuscript with other authors.

## Additional information

**Competing interests:** The authors declare no competing financial interests.

**Publisher's note**: 

