## [Peer Review File · Nature Communications]

Reviewers' Comments:

Reviewer #1 (Remarks to the Author):

The loading and activation of the Mcm2-7 replicative helicase is the central regulatory event that couples cell cycle progression to DNA replication. Despite the overall importance of this process, the mechanistic details surrounding this event have proven difficult to uncover. As Mcm2-7 forms a closed ring structure in isolation, one long-standing mystery of the process is how this ring is opened so that DNA can load into the central channel. Toward this end, this manuscript describes the physical interaction between Mcm2-7 and Cdt1, an essential protein required in vivo to load Mcm2-7 onto origins of replication. By combining protein interaction experiments with EM-based structure determination, the authors identify the physical contacts between Cdt1 and Mcm2-7. Intriguingly, the authors demonstrate that Cdt1 stabilizes a specific conformation of Mcm2-7 containing a discontinuity in the ring (gate-open form), suggesting a plausible mechanism for DNA loading in the Mcm central channel through this discontinuity.

Big issues:

In general, the thesis that Cdt1 stabilizes the gate-open form of Mcm2-7 is convincing and the main body of the manuscript is well-written. However, one shortcoming of the manuscript regards the underlying mechanism behind this structure. Two non-exclusive mechanisms are implied at different points in the text: 1) that among a population of Mcm2-7 molecules that interconvert between the gate-open and gate-closed forms, Cdt1 specifically binds and stabilizes the gate-open form, thereby shifting the equilibrium in favor of this Mcm2-7 conformation, or 2) that the interconversion between the different Mcm2-7 conformations requires ATP hydrolysis at some/many of the six Mcm ATPase active sites, and that Cdt1 stabilizes the gate-open conformation by regulating the ATP hydrolysis of these site(s). Although the authors provide evidence that Cdt1 is able to inhibit bulk ATP hydrolysis of at least some of the Mcm ATPase active sites (Fig. 1D), And that Cdt1 may dissociate the Mcm6/2/5/3 ‘tetramer (more below), the connection between Mcm ATP hydrolysis and Cdt1 function in Mcm2-7 loading is lacking.

In short, Fig. 1D raises more problems than it solves. To summarize a much longer draft of this review, I recommend that the authors fix this dilemma in one of two ways: 1) provide a comprehensive analysis of the effects of Cdt1 on Mcm2-7 ATP hydrolysis with the goal of elucidating this such interaction assists ring opening, or 2) remove Fig. 1D, leave the involvement of Mm ATP hydrolysis an open issue for this manuscript, and pursue this problem in a separate paper. Given the complexity of the experimental system, plan #2 seems more appropriate.

Smaller issues:

1. Main text.

a. Mass spectrophotometer analysis was used to assign Mcm subunit identify in the crosslinking experiments shown in Fig. 1b, but no details of this analysis are provided in either the M&M or Supplemental. Please remedy.

b. Fig. 1F and associated Supplemental figures depend upon analysis of the Mcm6/2/3/5 tetramer to provide evidence that Cdt1 is able to open the Mcm gate. However, previous independent studies (Megan Davey and Tony Schwacha) were unable to observe physical association between isolated Mcm2 and Mcm5 subunits, and inferred that Mcm2/5 association is only promoted by the structure of the full hexamer. These observations cast doubt on the ability to form the Mcm6/2/5/3 “tetramer”, and no data is presented to confirm that these subunits actually tetramerize, as opposed to being present as two unassociated dimers of roughly equal size that co-migrate during gel filtration. Lack of initial tetramerization would completely change the interpretation of the presented experiments; please provide appropriate control data.

c. Fig. 2. This Figure and the corresponding text have various problems. The choice of structures presented seems a bit odd. This figure contains 3 Cdt1 substructures that were used to assemble the full Cdt1 structure (Sup. Fig 2). Why not present the full length composite in Fig. 2 and relegate the substructures to Supplemental? I appreciate that there is more uncertainty in the full structure than the substructures, but as long as you make this point clear, I don't see a problem with presenting the full structure in the main paper. Also, the C-terminal Cdt1 substructure in Fig 2 lists amino acids 495-604 in text, but 493-604 in the figure. In the main text, it would be more useful to split the current description of this Figure (pages 5-6 in the

manuscript) into 2 paragraphs, with the basic results of the substructures (including the C-terminal structure that only appears as an afterthought at the end of this section) in the first paragraph, and the observations and supporting text concerning the structural homology between Cdt1 and various dioxygenases in the second paragraph. Finally, although the process of determining the N- and C-terminal Cdt1 structures are reasonably well documented in M&M, no related information is provided for the Cdt1 middle domain – clearly an oversight?

d. Fig. 3. What is the source of Mcm hexamers (reconstituted for *E. coli* or co-purified from yeast) in the loading assay? Although Frigola 2013 is cited for this assay, the 2013 paper is rather vague on this detail as well. As Cdt1 is ~ 68 kDa, and would be expected to be observed on the gel in Fig. 3A, why is it not observed in the input lanes? Critically, as the word “DNA” is not found in either the text or M&M regarding this experiment, the naïve reader has no idea why the Mcms are actually being “retained” in the high salt wash. Please provide appropriate details so readers who are not replication gurus may understand the assay.

e. EM issues. The central premise of this paper is that Cdt1, by an unspecified mechanism, preferentially associates with gate-open MCM hexamers. Toward this end, the authors present a limited number of class averages in Fig. 4, which clearly supports their desired point. However, making such a claim is not quite so simple for at least several reasons: 1) The possibility of cherry-picking the class averages. Typically, dozens to hundreds of class averages are generated at this step in the process. Although I don't quite understand the presented class averages for the wildtype Mcm2-7 in the absence of ATP (Sup Fig 3 only shows 5 class average – where are the rest?), the other structures (Sup. Figs 4-6) show at least 50 class averages. To the naïve reader, it is difficult in most cases to cleanly categorize any particular ring-forming class average as either “closed” or “open” using eyeball examination. Based on the full set of class averages shown in the Supplemental figures, the notion that Cdt1 associates with open rings is not so obvious. 2) The possibility of cherry-picking structures. In the end, researcher-guided class averages are used to assemble a final 3D structure. Typically, as even good protein preps have a certain number of misbehaving particulars, not all class averages or particles are used in the final structure. Has the final structure been biased due to the choice of particles used?

The above problems have nothing to do with the authors and their analysis, but are inherent with this methodology and are exasperated by the problem that Mcm2-7 forms a mixture of conformations. This reviewer is unaware of any simple way to completely eliminate these concerns. However, instead of sweeping this problem under the rug, more transparency is needed; one should note the limits of this analysis, and at least mention in M&M the final number of particles used in the analysis and provide the criteria that lead to their inclusion in the

final model so that the reader can form their own judgment. Alternatively, one might be able to use old-fashion biochemistry to better substantiate the claim that Cdt1 binds open rings; In the presence of ATP γ S, Cdt1/Mcm2-7 would be predicted to bind circular single stranded DNA much better than Mcm2-7 (which under these conditions has a closed ring structure), an assay previously used to assess Mcm gate conformation (Mol. Cell 2008 31:287).

f. In addition, with structural studies, isn't it customary to mention the resolution in the body of the paper? As nearly as I can tell from the various FSCs in supplemental, the EM structures are at ~ the 25 Angstrom level. Although this resolution pales relative to that attainable using recent advances in cryo-EM, this is excellent for negative staining and is of sufficient resolution to support the points presented in this manuscript.

2. Figure legends. Figure legends should generally be interpretable on their own without having to carefully read the Results or M&M section. Using this criterion, the current Figure legends are extremely wanting. Problems include but are not restricted to:

a. Sup Fig 1A- Sedimentation details? B – Essentially uninterpretable as presented. For example, what is the actual relationship between panels i and ii? One might for example imagine that lane 2 (i) might correspond to the second lane in ii (M3), but the mcm3 subunit in i does not seem to be GFP-tagged. Perhaps re-label the lanes in i M2-M7 if this is the intention? Panel D – legend and panel doesn't match (Mcm6DC used in both i and ii). Panel E – Controls? Where do the separate species run? Panel F – Protein concentrations?

b. Fig 1E legend: "Mapping domains of Cdt1 (i) interacting with Mcm2 and Mcm6 (ii)". I have no idea what was actually done to generate Fig. 1E.

3. Materials and Methods. Although citing assay references to shrink the length of the M&M is appropriate and recommended, one still needs to provide at least a brief description of relevant assays in either the M&M or the Results section so that the reader knows how they work (e.g., see 1D above). Such descriptions are generally lacking. In addition, to provide critical information both to evaluate the data as well as to repeat it, one needs to list the critical experimental parameters (e.g., protein and ATP concentrations in biochemical assays) in either the M&M or relevant Figure legends. Please provide such details. In addition

a. The M&M is largely dedicated to protein production details. Although important, for the most part these details are not crucial for understanding or evaluating the presented data. Much of this could perhaps be moved to the Supplemental. In addition, considerable information is

provided in a series of slender paragraphs listing previously purified proteins; this information might be better consolidated into a table.

b. Various proteins listed in Materials and Methods are labelled with nomenclature such as “JF12”. Very similar notation is used to define the oligos, plasmids, and yeast strains. I infer that the “JF12”-type nomenclature is Frigola lab shorthand for protein preparations isolated from a particular strain with this designation. This nomenclature, however, is completely meaningless to everyone else. What are the actual identities of these proteins? – Please fix.

4. References. At least several references poorly correspond to what is cited in the text. For example, reference #10, used to cite the discovery of the Mcm gate, is listed as Bochman et al 2008 MCB 28:5865. The relevant reference, I believe, is actually Bochman et. al., 2008 Mol. Cell 31:287.

Reviewer #2 (Remarks to the Author):

In this manuscript, Diffley, Costa and colleagues address the structure of Cdt1 and the consequences of its interactions with Mcm2-7. The authors demonstrate that Cdt1 interacts with three Mcm subunits with the strongest interaction with Mcm6 and progressively weaker interactions with Mcm2 and Mcm4. Consistent with previous structural studies, these interactions with Mcm2 and Mcm6 are mediated by the two winged-helix domains found in the middle and C-terminal regions of Cdt1. The authors provide additional data suggesting these interactions modulate Mcm2-7 ATPase activity and interactions between the Mcm2 and Mcm5 subunits. The authors determine the X-ray crystal structure of two regions of Cdt1, revealing the expected winged-helix domains and the structure of a previously undetermined N-terminal region related to DNA/RNA alkylators (although the enzymatic aspects of this fold are missing). The authors then analyze the consequences of deletion of the Cdt1 N-terminus, observing a defect in loading. The most important data in the manuscript are low-resolution EM-based structural studies showing that the Cdt1 bound complex is in a cracked-ring conformation, suggesting a key role of Cdt1 is to hold the Mcm2-7 ring in an open state. The authors conclude with a model based on this idea.

The EM studies provide data supporting the hypothesis that Cdt1 binding-release modulates the status of the Mcm2-5 gate. This is further supported by the biochemistry in Fig. 1F, although this data is less compelling given the partial nature of this Mcm complex involved. One question that should be addressed, is whether there is room for the Cdt1 middle domain to bind in the context of the closed Mcm ring? In addition, the authors should address whether there is room for dsDNA to pass between Mcm2-Mcm5 in the structures they have determined. The former would address whether loss of Cdt1 would be required to achieve the closed ring state and the latter would help to understand if Cdt1 binding is sufficient to hold the Mcm2-7 ring open during DNA entry into the central channel (as opposed to binding to ORC/Cdc6 facilitating this event).

The X-ray structural studies provide new insights into the structure of the N-terminal domain of Cdt1. One concern are the conclusions made based on the corresponding biochemical analyses of Cdt1 lacking this region. Consistent with previous studies, the authors observe reduced Mcm2-7 loading with this mutation. They then modify their loading assays to overcome this defect and test the resulting loaded Mcm2-7 in a S-phase extract assay and find no defect in generation of replication products. Their conclusion is that there is a non-essential role of the N-terminus in loading and no effect in the subsequent initiation and elongation reaction. There are two problems with these conclusions. First, genetic studies show that this mutant is lethal, so there must be an essential role of this domain. Second, although the authors assume that the altered loading assay only increases the amount of Mcm2-7 loaded by the mutant Cdt1, there is no evidence that this higher concentration/longer loading reaction also allows other defects in the resulting loaded Mcm2-7 complexes to be compensated for (such a defect in downstream events). In the end, it would be better for the authors to just conclude that the most likely defect that causes lethality is in loading since it would require substantially more data to demonstrate the conclusions that they currently make.

Overall this is an interesting set of observations that adds to our understanding of the mechanism of helicase loading and merits publication after revision.

Specific Points:

1. The authors should refer to the previous observations from Liang and colleagues indicating that Cdt1 bound to Mcm6 (Wu, R., Wang, J., & Liang, C. (2012). *Journal of Cell Science*, 125(Pt 1), 209–219. <http://doi.org/10.1242/jcs.094169>).

2. Based on the figure legend, the data in figure 3A does not contain any ATPγS experiments as described in the text. This experiment should be added to the manuscript.

3. The last sentence in the last paragraph before the discussion states that “the AAA+ domain of Mcm2 moves in close proximity to Mcm5, reconstituting a key active site that is essential for the ATPase function of MCM”. The authors should include a reference for this statement.

This last part of this same sentence is unclear but possibly very important (and may address the question raised above as to whether Cdt1 binding is compatible with the loaded form of Mcm2-7). The authors should explicitly state the evidence that the closed-ring form of Mcm2-7 disrupts the Cdt1 binding site. This be illustrated with structural images rather than the cartoon in Fig. 5.

4. The model suggests that ATP hydrolysis causes Cdt1 release which in turn causes ring closure. The authors should also include the possibility that ATP hydrolysis drives ring closure which in turn causes Cdt1 release. The data presented cannot distinguish between these two possibilities and the latter possibility might fit better with the strong role of ATPases that don't interact with Cdt1 in helicase loading.

5. It seems inappropriate to end the abstract with a very explicit model that is not the only one possible at this point. It would be better to shorten this to discuss the likely role of Cdt1 in holding the ring in an open position that is most strongly supported by the current data.

Reviewer #3 (Remarks to the Author):

Frigola et al. used biochemical and structural approaches to propose a model on how Cdt1 regulates the opening and closing of the MCM gate for DNA entry into the central channel of the hetero-hexameric assembly. Using biochemical assays, the authors demonstrate that Cdt1 interacts with MCM and more specifically that the N-terminus of Cdt1 is involved in MCM loading. Using x-ray crystallography, the authors determined the structures of N-, M-, and C-domains of Cdt1 separately. Finally, the authors resolved different states of the Cdt1-MCM

complexes in the presence or absence of ATP γ S using single particle analysis and negative-staining EM.

The manuscript is well written and the results contribute to the understanding of how DNA is loaded on the MCM during DNA replication in eukaryotes. One main criticism of the work is the resolution of the reconstructions and the method of choice for resolving the Cdt1-MCM complex, particularly with negative staining and not cryo-EM, which poses limits on fitting of atomic coordinates and hampers the interpretation of the maps. While it is understandable that protein assemblies are more prone to aggregation and complex disassembly during freezing for cryo-EM studies, recent work by the same authors (Ali et al., Nat Comm., 2016) for a similar complex (Cdc45-MCM-GINS) suggest such studies are possible.

Specific points:

- In Suppl. fig 7, it not obvious how the authors established the handedness of their reconstructions as both fits have remaining unoccupied EM-densities. The authors should provide a movie or different views for the reviewer to better evaluate the two fits shown in this figure.

- The 3D classification scheme is not explained in either the respective figure legend and/or in the Material and Methods and should be further elaborated on. For example, why did the authors choose to combine classes 1 and 2 in the Apo Cd10-MCM dataset? The two classes are clearly not the same and show different degrees of gate opening. Similarly, in the ATP γ S dataset, the 3D classes are also different but they were later combined and refined as one class.

- The distortion in the hetero-hexameric rings in the 3D reconstruction of the ATP γ S-bound state of MCM is a bit surprising. The flat views of the rings in the 2D classification, as shown in Suppl. fig 4, do not contain such distortion. Did the authors classify for these states? Also the 3D classes look very heterogeneous. Could this be due to a staining artifact? One suggestion that the authors can try is incubating with different nucleotides (i.e. GDP or GTP) and evaluate semi-quantitatively the ring conformation on the level of 2D classes. This approach will strengthen the functional relevance to the current observed state. This will also rule out variations due to staining, which is a common artifact when doing negative-staining EM.

- How did the authors validate their 3D reconstructions? At this resolution, a side-by-side comparison of the re-projections of the final reconstruction against reference-free 2D class averages is very helpful. This would allow for a better evaluation of the quality of the 3D reconstruction; particularly the lobes of the hetero-hexameric rings look sharper in the 2D classes versus the 3D reconstructions where they are smeared out.

- The fit of Cdt1 as shown in Fig. 4 and Suppl. fig. 8 is not clear and it is difficult to validate due to the current resolution of the map. The authors should be more careful with the interpretation of such densities. Also, how did the authors segment their maps? The approach used should be described in the Materials and Methods for more clarification.

Minor points:

- The authors could also show the chromatograms of their gel-filtration analysis in Suppl. fig 1, panel G, for the readers to evaluate the quality of the peaks.

- As a good practice, the authors should show representative areas of the electron density maps of their x-ray structures in Supplementary Data, particularly for the areas where they observe differences with super-imposed x-ray structures shown in this study.

- What is panel E, in Fig 4? It is not in the figure legend.

Response to Reviewers:

Reviewer #1:

We were pleased reviewer 1 thought our paper was "...convincing and ...well-written". This reviewer raised one 'Big issue':

Two non-exclusive mechanisms are implied at different points in the text: 1) that among a population of Mcm2-7 molecules that interconvert between the gate-open and gate-closed forms, Cdt1 specifically binds and stabilizes the gate-open form, thereby shifting the equilibrium in favor of this Mcm2-7 conformation, or 2) that the interconversion between the different Mcm2-7 conformations requires ATP hydrolysis at some/many of the six Mcm ATPase active sites, and that Cdt1 stabilizes the gate-open conformation by regulating the ATP hydrolysis of these site(s). Although the authors provide evidence that Cdt1 is able to inhibit bulk ATP hydrolysis of at least some of the Mcm ATPase active sites (Fig. 1D), And that Cdt1 may dissociate the Mcm6/2/5/3 "tetramer (more below), the connection between Mcm ATP hydrolysis and Cdt1 function in Mcm2-7 loading is lacking.

This is an important issue. We realise now that several passages in the original text juxtaposed inhibition of ATP hydrolysis and MCM ring opening in a way that sounded causal – this was never our intention. We completely agree with this reviewer that none of our data support a model in which regulation of ATP hydrolysis by Cdt1 stabilizes the gate-open conformation. In fact, all of our data are consistent with the idea that MCM interconverts freely between the open and closed forms (from EM analysis of MCM alone), and Cdt1 binds and shifts the equilibrium toward the open conformation. Our data are more consistent with the idea that ATPase inhibition is a consequence of MCM being held in the open conformation, not a cause of the open configuration. Indeed, if simply inhibiting ATP hydrolysis caused ring opening, ATPγS should put the MCM complex without Cdt1 into the open configuration; it is, instead, pushed into the closed conformation. We have re-written the text in several places to make this clear.

Smaller issues:

1. Main text.

a. Mass spectrophotometer analysis was used to assign Mcm subunit identify in the crosslinking experiments shown in Fig. 1b, but no details of this analysis are provided in either the M&M or Supplemental. Please remedy.

The cross-linked bands corresponding to 'M6 + Cdt1' and 'M2 + Cdt1' were initially identified by immunoblot (Supp. Fig. 1C); however, the lowest of the cross-linked bands did not contain either Mcm2 or Mcm6 by immunoblot. Mass spectrometry showed that this band was over-represented in peptides from Mcm4, hence our assignment. Mass spec also confirmed enrichment of Mcm2 and 6 in the other bands. We have included an Excel file with the Mass spec data from the bands we examined and we have re-written the description of these experiments to clarify this point.

b. Fig. 1F and associated Supplemental figures depend upon analysis of the Mcm6/2/3/5 tetramer to provide evidence that Cdt1 is able to open the Mcm gate. However, previous independent studies (Megan Davey and Tony Schwacha) were unable to observe physical association between isolated Mcm2 and Mcm5 subunits, and inferred that Mcm2/5 association is only promoted by the structure of the full hexamer. These observations cast doubt on the ability to form the Mcm6/2/5/3 "tetramer", and no data is presented to confirm that these subunits actually tetramerize, as opposed to being present as two unassociated dimers of roughly equal size that co-migrate during gel filtration. Lack of initial tetramerization would completely change the interpretation of the presented experiments; please provide appropriate control data.

It is true that Mcm2/5 interactions were not detected by Davey et al. or Bochman et al. However, these experiments were all performed in 0.1M chloride (NaCl and KCl respectively), which we now know disrupts the MCM complex. Our experiments were done in potassium glutamate buffer, which stabilises MCM subunit interactions. The elution fractions on this analytical gel filtration are highly reproducible. As shown in Supp. Fig. 1g, addition of Cdt1 causes not only causes Mcm2/6 to elute 1-2 fractions earlier, with Cdt1, but also causes Mcm3/5 to elute 1-2 fractions later. This shouldn't happen if 3/5 and 4/6 are two unassociated dimers. Moreover, as we show in Fig. 1, Mcm10 can pull down Mcm3 and 5 despite the fact that it does not interact with either subunit (Douglas and Diffley 2015). Therefore, two different biochemical approaches indicate that Cdt1 destabilises the 2/5 interface which,

of course, is also consistent with the EM data. We think the biochemistry adds something important and unique to this paper. We have also added more detail about how these experiments were done in Experimental Procedures.

c. Fig. 2. This Figure and the corresponding text have various problems. The choice of structures presented seems a bit odd. This figure contains 3 Cdt1 substructures that were used to assemble the full Cdt1 structure (Sup. Fig 2). Why not present the full length composite in Fig. 2 and relegate the substructures to Supplemental? I appreciate that there is more uncertainty in the full structure than the substructures, but as long as you make this point clear, I don't see a problem with presenting the full structure in the main paper.

Figure 2 shows the crystal structures of the 2 portions of Cdt1 (i) the N-terminal domain and middle domain and (ii) the C-terminal domain. The domains are colour coded and dotted lines represent linkers which are not observed in our crystal structures. Together, Figure 2 represents all of the Cdt1 structure that has been crystallographically determined. Additionally, in Supplementary Fig. 10 we now include the atomic docking of the crystallographic structures into the MCM-Cdt1 cryo-EM structure by Zhai et al. 2017. Importantly, the resulting atomic model differs from their model in that the N-terminal dioxygenase fold of Cdt1 is ~180 degrees rotated in the Gao structure and the connectivity to the MD domain of Cdt1 is inverted. This issue arises because NTD-Cdt1 is pseudo-two-fold symmetric when visualized at ~7 Å resolution. Our crystallographic data hence correct an error in the published MCM-Cdt1 atomic model (detailed in Supplementary Fig. 10).

Also, the C-terminal Cdt1 substructure in Fig 2 lists amino acids 495-604 in text, but 493-604 in the figure.

We thank the reviewer for pointing out this discrepancy. Now corrected in the figure.

In the main text, it would be more useful to split the current description of this Figure (pages 5-6 in the manuscript) into 2 paragraphs, with the basic results of the substructures (including the C-terminal structure that only appears as an afterthought at the end of this section) in the first paragraph, and the observations and supporting text concerning the structural homology between Cdt1 and various dioxygenases in the second paragraph. Finally, although the process of determining the N- and C-terminal Cdt1 structures are reasonably well documented in M&M, no related information is provided for the Cdt1 middle domain – clearly an oversight?

Crystal structures of two portions of Cdt1 are described in this paper – (i) the N-terminal region which contains both the N-terminal and middle domain (there are 2 crystal forms of this) and (ii) the C-terminal domain. Together, these structures describe the majority of Cdt1.

Additionally, during the course of this study, the crystal structure of the isolated middle domain was also solved. This structure became redundant upon solving the N-terminal and middle domain portion of Cdt1, nevertheless we have also deposited this structure with the pdb.

d. Fig. 3. What is the source of Mcm hexamers (reconstituted for E. coli or co-purified from yeast) in the loading assay? Although Frigola 2013 is cited for this assay, the 2013 paper is rather vague on this detail as well.

The source of the proteins has now been described in more detail in Experimental Procedures.

As Cdt1 is ~ 68 kDal, and would be expected to be observed on the gel in Fig. 3A, why is it not observed in the input lanes?

The experiments in Fig. 3A were all performed in ATP. Under these conditions, Cdt1 is released; hence it is not visible. Unfortunately, I mistakenly described this experiment as being performed in ATPγS. I apologise for this mistake. We note that Fig. 3a lane 5 shows that even in ATP the recruitment of MCM by the truncated Cdt1 after low salt wash appears to be as efficient as the full length Cdt1 (lanes 1 and 3).

Critically, as the word “DNA” is not found in either the text or M&M regarding this experiment, the naïve reader has no idea why the Mcms are actually being “retained” in the high salt wash. Please provide appropriate details so readers who are not replication gurus may understand the assay.

This issue has been addressed by adding more detail in the main section of the manuscript as well as the Experimental Procedures.

e. EM issues. The central premise of this paper is that Cdt1, by an unspecified mechanism, preferentially associates with gate-open MCM hexamers.

Toward this end, the authors present a limited number of class averages in Fig. 4, which clearly supports their desired point. However, making such a claim is not quite so simple for at least several reasons: 1) The possibility of cherry-picking the class averages. Typically, dozens to hundreds of class averages are generated at this step in the process. Although I don't quite understand the presented class averages for the wildtype Mcm2-7 in the absence of ATP (Sup Fig 3 only shows 5 class average – where are the rest?),

We have now included a larger gallery of class averages for apo MCM in Supplementary Fig. 4.

the other structures (Sup. Figs 4-6) show at least 50 class averages. To the naïve reader, it is difficult in most cases to cleanly categorize any particular ring-forming class average as either “closed” or “open” using eyeball examination. Based on the full set of class averages shown in the

Supplemental figures, the notion that Cdt1 associates with open rings is not so obvious. 2) The possibility of cherry-picking structures. In the end, researcher-guided class averages are used to assemble a final 3D structure. Typically, as even good protein preps have a certain number of misbehaving particulars, not all class averages or particles are used in the final structure. Has the final structure been biased due to the choice of particles used?

While open rings can be clearly recognized in 2D averages, it is not possible, formally, to distinguish closed rings and tilted open rings in two dimensions. We have modified the text to acknowledge this caveat. Importantly, three-dimensional structures have not been determined using particles contributing to class averages based on their (open or closed) features. Employing a commonly accepted protocol, instead, all particles contributing to sharp, featured class averages were included in the pool of particles used for three dimensional classification and refinement. In other words, the open or closed state of the particles was determined after inspection of the resulting 3D structures, while the 2D data only provides independent support to the conclusion drawn from 3D refinement work. We have modified the text to clarify this important point.

The above problems have nothing to do with the authors and their analysis, but are inherent with this methodology and are exasperated by the problem that Mcm2-7 forms a mixture of conformations. This reviewer is unaware of any simple way to completely eliminate these concerns. However, instead of sweeping this problem under the rug, more transparency is needed; one should note the limits of this analysis, and at least mention in M&M the final number of particles used in the analysis and provide the criteria that lead to their inclusion in the final model so that the reader can form their own judgment.

We have now modified the results section to acknowledge the limitations of the negative stain EM technique for structural analysis. The final number of particles used in the analysis is now included in Supplementary figure 11, where the 3D classification/refinement procedure is detailed.

Alternatively, one might be able to use old-fashion biochemistry to better substantiate the claim that Cdt1 binds open rings; In the presence of ATP γ S, Cdt1/Mcm2-7 would be predicted to bind circular single stranded DNA much better than Mcm2-7 (which under these conditions has a closed ring structure), an assay previously used to assess Mcm gate conformation (Mol. Cell 2008

31:287).

This is an interesting idea, however, Cdt1-MCM does not bind appreciably to DNA on its own. We presume because Cdt1 – by holding the ring open – prevents topological binding of MCM to DNA.

f. In addition, with structural studies, isn't it customary to mention the resolution in the body of the paper? As nearly as I can tell from the various FSCs in supplemental, the EM structures are at ~ the 25 Angstrom level. Although this resolution pales relative to that attainable using recent advances in cryo-EM, this is excellent for negative staining and is of sufficient resolution to support the points presented in this manuscript.

Resolution of the negative stain EM structures is now mentioned in the Results section.

2. Figure legends. Figure legends should generally be interpretable on their own without having to carefully read the Results or M&M section. Using this criterion, the current Figure legends are extremely wanting. Problems include but are not restricted to:

a. Sup Fig 1A- Sedimentation details? B – Essentially uninterpretable as presented. For example, what is the actual relationship between panels i and ii? One might for example imagine that lane 2 (i) might correspond to the second lane in ii (M3), but the mcm3 subunit in i

does not seem to be GFP-tagged. Perhaps re-label the lanes in i M2-M7 if this is the intention? Panel D – legend and panel doesn't match (Mcm6DC used in both i and ii). Panel E – Controls? Where do the separate species run? Panel F – Protein concentrations?

We have amended figures legends to Fig.1 and Supp. 1.

b. Fig 1E legend: "Mapping domains of Cdt1 (i) interacting with Mcm2 and Mcm6 (ii)". I have no idea what was actually done to generate Fig. 1E.

We have amended figures legends to Fig.1 and Supp. 1.

3. Materials and Methods. Although citing assay references to shrink the length of the M&M is appropriate and recommended, one still needs to provide at least a brief description of relevant assays in either the M&M or the Results section so that the reader knows how they work (e.g., see 1D above). Such descriptions are generally lacking. In addition, to provide critical information both to evaluate the data as well as to repeat it, one needs to list the critical experimental parameters (e.g., protein and ATP concentrations in biochemical assays) in either the M&M or relevant Figure legends. Please provide such details. In addition:

a. The M&M is largely dedicated to protein production details. Although important, for the most part these details are not crucial for understanding or evaluating the presented data. Much of this could perhaps be moved to the Supplemental. In addition, considerable information is provided in a series of slender paragraphs listing previously purified proteins; this information might be better consolidated into a table.

Experimental Procedures have been rewritten. Firstly, more technical details of the assays are added and secondly, following this suggestion, we have moved the cloning and purification of the proteins to Supplementary Material. We think a textual description allows for more detail than a table.

b. Various proteins listed in Materials and Methods are labelled with nomenclature such as "JF12". Very similar notation is used to define the oligos, plasmids, and yeast strains. I infer that the "JF12"-type nomenclature is Frigola lab shorthand for protein preparations isolated from a particular strain with this designation. This nomenclature, however, is completely meaningless to everyone else. What are the actual identities of these proteins? – Please fix.

We have revised the text so every yeast strain has a 'y' before the JF to differentiate them from the primers. Wherever feasible we used more descriptive names in place of yJF (e.g. MCM-Cdt1 FLAG instead of yJF59).

4. References. At least several references poorly correspond to what is cited in the text. For example, reference #10, used to cite the discovery of the Mcm gate, is listed as Bochman et al 2008 MCB 28:5865. The relevant reference, I believe, is actually Bochman et. al., 2008 Mol. Cell 31:287.

Apologies for this EndNote problem. This reference is fixed and I believe the rest are correct.

Reviewer #2:

We were pleased this reviewer felt "Overall this is a interesting set of observations that adds to our understanding of the mechanism of helicase loading and merits publication after revision." We feel we have addressed all of his/her concerns.

In this manuscript, Diffley, Costa and colleagues address the structure of Cdt1 and the consequences of its interactions with Mcm2-7. The authors demonstrate that Cdt1 interacts with three Mcm subunits with the strongest interaction with Mcm6 and progressively weaker interactions with Mcm2 and Mcm4. Consistent with previous structural studies, these interactions with Mcm2 and Mcm6 are mediated by the two winged-helix domains found in the middle and C-terminal regions of Cdt1. The authors provide additional data suggesting these interactions modulate Mcm2-7 ATPase activity and interactions between the Mcm2 and Mcm5 subunits. The authors determine the X-ray crystal structure of two regions of Cdt1, revealing the expected winged-helix domains and the structure of a previously undetermined N-terminal region related to DNA/RNA alkylators (although the enzymatic aspects of this fold are missing). The authors then analyze the consequences of deletion of the Cdt1 N-terminus, observing a defect in loading. The most important data in the manuscript are low-resolution EM-based structural studies showing that the Cdt1 bound complex is in a cracked-ring conformation, suggesting a key role of Cdt1 is to hold the Mcm2-7 ring in an open state. The authors conclude with a model based on this idea.

The EM studies provide data supporting the hypothesis that Cdt1 binding-release modulates the status of the Mcm2-5 gate. This is further supported by the biochemistry in Fig. 1F, although this data is less compelling given the partial nature of this Mcm complex involved. One question that should be addressed, is whether there is room for the Cdt1 middle domain to bind in the context of the closed Mcm ring?

We now discuss the new high resolution OCCM structure (trapped in the ATP_γS state - Yuan et al. 2017), which indicates that Cdt1 binding is maintained when MCM is converted to a planar configuration upon ORC binding. The new cryo-EM OCCM helps us substantiate our proposal that Cdt1 eviction does not depend upon MCM planarization per se, but rather upon the ATPase reconfiguration observed after ATP hydrolysis (observed in the loaded, double hexameric MCM).

In addition, the authors should address whether there is room for dsDNA to pass between Mcm2-Mcm5 in the structures they have determined. The former would address whether loss of Cdt1 would be required to achieve the closed ring state and the latter would help to understand if Cdt1 binding is sufficient to hold the Mcm2-7 ring open during DNA entry into the central channel (as opposed to binding to ORC/Cdc6 facilitating this event).

Due to the low resolution of our structural data it is impossible to establish whether duplex DNA could easily thread through the Mcm2-5 opening in the MCM-Cdt1 complex. We can, however, now compare our structure to the high-resolution OCCM structure and conclude that duplex DNA is more likely to thread through the 2-5 gate in the MCM-Cdt1 state, than in the narrower opening observed in the planar, notched MCM ring observed in the OCCM. We have now modified our results section to include this observation. The interaction between Cdt1 MD domain and Mcm2 AAA+ ATPase domain (described in the new high-resolution OCCM structure) is lost in the post-catalytic double hexamer, likely weakening the MCM-Cdt1 interaction and, we propose, promoting Cdt1 eviction.

The X-ray structural studies provide new insights into the structure of the N-terminal domain of Cdt1. One concern are the conclusions made based on the corresponding biochemical analyses of Cdt1 lacking this region. Consistent with previous studies, the authors observe reduced Mcm2-7 loading with this mutation. They then modify their loading assays to overcome this defect and test the resulting loaded Mcm2-7 in a S-phase extract assay and find no defect in generation of replication products. Their conclusion is that there is a non-essential role of the N-terminus in loading and no effect in the subsequent initiation and elongation reaction. There are two problems with these conclusions. First, genetic studies show that this mutant is lethal, so there must be an essential role of this domain. Second, although the authors assume that the altered loading assay only increases the amount of Mcm2-7 loaded by the mutant Cdt1, there is no evidence that this higher concentration/longer loading reaction also allows other defects in the resulting loaded Mcm2-7 complexes to be compensated for (such a defect in downstream events). In the end, it would be better for the authors to just conclude that the most likely defect that causes lethality is in loading since it would require substantially more data to demonstrate the conclusions that they currently make.

We have extensively re-written this section to make clear exactly what we were testing and what our results tell us. After seeing that the N-terminus had an enzymatic fold, we were reminded of work from Takara and Bell showing that deletion of this domain resulted in reduced MCM loading, and that MCM loaded with this deletion mutant was poorly replicated in extracts. Indeed we were able to reproduce this defect. Because the defect in replication seemed to be greater than the defect in loading, Takara and Bell concluded that the N-terminus was important for MCM loading, but also played a second role downstream of MCM loading in replication. Given the presence of the dioxygenase fold, this was intriguing. However, there was a potentially trivial explanation for the Takara and Bell results: we have found that replication in extracts is non-linear with respect to input MCM, and a small defect in loading might cause a greater defect in replication. We sought to clarify this by eliminating the difference in loading. Unfortunately, it turns out that when we normalized for MCM loading, there was no difference in replication (Figure 3b,d). Thus, we see no evidence for a downstream role in replication for Cdt1. I recognize that bits of this section were confusing and didn't distinguish essentiality *in vivo* and *in vitro*. The conclusion has now been changed to make it clear that the essentiality of the N-terminus *in vivo* likely reflects the relatively severe defect in MCM loading we see *in vitro*.

Specific Points:

1. The authors should refer to the previous observations from Liang and colleagues indicating that Cdt1 bound to Mcm6 (Wu, R., Wang, J., & Liang, C. (2012). *Journal of Cell Science*, 125(Pt 1), 209–219. <http://doi.org/10.1242/jcs.094169>).

This reference has now been included.

2. Based on the figure legend, the data in figure 3A does not contain any ATPgS experiments as described in the text. This experiment should be added to the manuscript.

I apologise for the confusion regarding this experiment – as described above in our response to reviewer 1 this experiment was performed in ATP, not ATPγS. This has now been clarified. We have not included an ATPγS experiment because recruitment of MCM doesn't require Cdt1 (Frigola et al. 2012). Moreover, we note that Fig. 3a lane 5 shows that even in ATP the recruitment of MCM after low salt wash by the truncated appears to be as efficient as the full length Cdt1 (lanes 1 and 3).

3. The last sentence in the last paragraph before the discussion states that “the AAA+ domain of Mcm2 moves in close proximity to Mcm5, reconstituting a key active site that is essential for the ATPase function of MCM”. The authors should include a reference for this statement.

Done (Li et al. *Nature* 2015).

This last part of this same sentence is unclear but possibly very important (and may address the question raised above as to whether Cdt1 binding is compatible with the loaded form of Mcm2-7). The authors should explicitly state the evidence that the closed-ring form of Mcm2-7 disrupts the Cdt1 binding site. This be illustrated with structural images rather than the cartoon in Fig. 5.

We have now modified the Discussion section to clarify this important point. We have decided to keep the ball-and-stick depiction to describe the structural transitions between MCM-Cdt1, OCCM and loaded double hexamer, because we deemed this representation more intelligible than the atomic structure representation. Importantly, this ‘cartoon’ was obtained by identifying the center of mass of each NTD and ATPase domains in the MCM and therefore represents a rigorous structural representation of the MCM complex during loading. We have modified the Discussion and Figure caption to clarify this important point.

4. The model suggests that ATP hydrolysis causes Cdt1 release which in turn causes ring closure. The authors should also include the possibility that ATP hydrolysis drives ring closure which in turn causes Cdt1 release. The data presented cannot distinguish between these two possibilities and the latter possibility might fit better with the strong role of ATPases that don't interact with Cdt1 in helicase loading.

We have modified the discussion section to acknowledge both possibilities; we provide a detailed discussion of how the ATP hydrolysis and Cdt1 release might be coupled.

5. It seems inappropriate to end the abstract with a very explicit model that is not the only one possible at this point. It would be better to shorten this to discuss the likely role of Cdt1 in holding the ring in an open position that is most strongly supported by the current data.

We have re-written this last sentence to clarify fact from proposal, and we have toned down our proposal.

Reviewer #3:

We were pleased this reviewer felt “The manuscript is well written and the results contribute to the understanding of how DNA is loaded on the MCM during DNA replication in eukaryotes.”

Frigola et al. used biochemical and structural approaches to propose a model on how Cdt1 regulates the opening and closing of the MCM gate for DNA entry into the central channel of the hetero-hexameric assembly. Using biochemical assays, the authors demonstrate that Cdt1 interacts with MCM and more specifically that the N-terminus of Cdt1 is involved in MCM loading. Using x-ray crystallography, the authors determined the structures of N-, M-, and C-domains of Cdt1 separately. Finally, the authors resolved different states of the Cdt1-MCM complexes in the presence or absence of ATPγS using single particle analysis and negative-staining EM.

The manuscript is well written and the results contribute to the understanding of how DNA is loaded on the MCM during DNA replication in eukaryotes. One main criticism of the work is the resolution of the reconstructions and the method of choice for resolving the Cdt1-MCM complex, particularly with negative staining and not cryo-EM, which poses limits on fitting of atomic coordinates and hampers the interpretation of the maps. While it is understandable that protein

assemblies are more prone to aggregation and complex disassembly during freezing for cryo-EM studies, recent work by the same authors (Ali et al., Nat Comm., 2016) for a similar complex (Cdc45-MCM-GINS) suggest such studies are possible.

We recognize that the use of cryo-EM would have provided a more detailed structure. Our data are however of sufficient resolution to describe the gross reconfiguration of the MCM ring that are required in the early steps of helicase loading. We also provide considerable biochemical and X-ray crystallographic data to support our model.

Specific points:

- *In Suppl. fig 7, it not obvious how the authors established the handedness of their reconstructions as both fits have remaining unoccupied EM-densities. The authors should provide a movie or different views for the reviewer to better evaluate the two fits shown in this figure.*

We now include a new video (Supplementary Movie 1), explaining how the MCM-Cdt1 molecular model was built. Markedly chiral features are recognizable, in particular in the NTD-MCM tier, which help establish the handedness of the EM map. Notably, a left- (rather than right-) handed hexameric spiral for the MCM agrees with the structure of *E. cuniculi* MCM and of *Drosophila* MCM, the latter established using Rosenthal & Henderson's free-hand test (Lyubimov et al. PNAS 2012).

- *The 3D classification scheme is not explained in either the respective figure legend and/or in the Material and Methods and should be further elaborated on.*

We have now included a description of the 3D classification scheme in the Supplementary Figure caption.

For example, why did the authors choose to combine classes 1 and 2 in the Apo Cd10-MCM dataset? The two classes are clearly not the same and show different degrees of gate opening. Similarly, in the ATP γ S dataset, the 3D classes are also different but they were later combined and refined as one class.

We have indeed attempted to refine a structure for the isolated class 1 and 2 in the apo Cdt1-MCM. However, the refined structures appear qualitatively poor. Much better results were obtained when combining classes 1 and 2. Although some differences appear in the two structures, the 3D classification and refinement results indicate that the MCM is an open spiral when Cdt1 bound (our only claim, based on this dataset). The same applies to the ATP γ S-MCM-Cdt1 dataset.

- *The distortion in the hetero-hexameric rings in the 3D reconstruction of the ATP γ S-bound state of MCM is a bit surprising. The flat views of the rings in the 2D classification, as shown in Suppl. fig 4, do not contain such distortion.*

The ATP γ S-MCM ring is closed in the 3D structure as shown in the reference-free 2D averages. We have now included a panel matching the re-projections of our 3D structure to the independently generated 2D averages, to address this important point. The extended C-terminal WH appendices found in Mcm3,4,5,6 might give the impression of a distorted rings, in the surface rendering of the ATP γ S-MCM map.

Did the authors classify for these states? Also the 3D classes look very heterogeneous. Could this be due to a staining artifact?

We acknowledge that the 3D classes appear heterogeneous, this is expected given that the open MCM ring is inherently less stable than a topologically closed hexameric ring assembly. The same conclusions (MCM complexed with Cdt1 is topologically open) have now been observed in a recent study (Zhai et al. 2017).

One suggestion that the authors can try is incubating with different nucleotides (i.e. GDP or GTP) and evaluate semi-quantitatively the ring conformation on the level of 2D classes. This approach will strengthen the functional relevance to the current observed state. This will also rule out variations due to staining, which is a common artifact when doing negative-staining EM.

We have performed an experiment similar to that suggested by this reviewer, imaging ATP treated MCM or MCM-Cdt1. Like in the apo state and unlike in the ATP γ S state, ATP-MCM exists as a mixture of open and closed rings. We speculate that this transition might be due to the fact that hydrolysis and nucleotide exchange promote the interconversion between the open and closed MCM conformers. Conversely, ATP-MCM-Cdt1 only showed open rings, clearly recognizable from the end-on views – this result was expected given that Cdt1 appears to select for an open form of MCM both in the apo and ATP γ S forms. Our new data are consistent with our conclusions, but do not add to them and introduce

an element of uncertainty given that we cannot determine the nucleotide state of the two ATP-incubated complexes. We prefer therefore not to include these new data in our revised work.

- How did the authors validate their 3D reconstructions? At this resolution, a side-by-side comparison of the re-projections of the final reconstruction against reference-free 2D class averages is very helpful. This would allow for a better evaluation of the quality of the 3D reconstruction; particularly the lobes of the hetero-hexameric rings look sharper in the 2D classes versus the 3D reconstructions where they are smeared out.

We have now included a match between reference-free 2D averages and 3D re-projections for every refined 3D structure included in this manuscript (Supplementary Figs 5, 6 and 7).

- The fit of Cdt1 as shown in Fig. 4 and Suppl. fig. 8 is not clear and it is difficult to validate due to the current resolution of the map. The authors should be more careful with the interpretation of such densities.

This is an important point. Since the first submission of our manuscript, two higher resolution structures containing MCM and Cdt1 (MCM-Cdt1 at ~7 Å resolution OCCM at near atomic resolution) have been published. For both structures an atomic model has been deposited to the PDB (with the local resolution for the NTD-Cdt1 density averaging ~7 Å in both cases), however the connectivity between NTD and MD domains is ~180° offset. To establish the relative orientation of NTD-Cdt1 and MCM, we have therefore used our NTD-MD Cdt1 structure for docking into both the MCM-Cdt1 and OCCM cryo-EM maps. This is how we can establish that the published MCM-Cdt1 structure contains an incorrect NTD-Cdt1 model (while the OCCM structure is correct). Importantly, our new atomic docking exercises reveal that the NTD-MD docking results originally obtained using the low resolution MCM-Cdt1 data are correct.

Also, how did the authors segment their maps? The approach used should be described in the Materials and Methods for more clarification.

We used both the *color zone* and *segment map* option implemented in UCSF Chimera. This is now detailed in the Materials and Methods.

Minor points:

- The authors could also show the chromatograms of their gel-filtration analysis in Suppl. fig 1, panel G, for the readers to evaluate the quality of the peaks.

The amounts of protein used in these experiments barely registered on the UV monitor. Moreover, given the relatively small changes in the behaviour of these mixed proteins, we feel gels are the best way of presenting these data.

- As a good practice, the authors should show representative areas of the electron density maps of their x-ray structures in Supplementary Data, particularly for the areas where they observe differences with super-imposed x-ray structures shown in this study.

These are now included in Supplementary Figure 2.

- What is panel E, in Fig 4? It is not in the figure legend.

We have now modified the Figure legend to describe Panel E.

Reviewers' Comments:

Reviewer #1 (Remarks to the Author):

The resubmitted manuscript "Cdt1 Stabilizes an Open MCM Ring for Helicase Loading" has successfully addressed Reviewer concerns.

Reviewer #2 (Remarks to the Author):

The manuscript by Diffley, Costa and colleagues is much improved and they have addressed all the previous important concerns. One minor point, the authors should mention recent studies from Ticau et al that support the authors supposition that there is a connection between Mcm2-7 ring closure and release of Cdt1 from Mcm2-7.

Reviewer #3 (Remarks to the Author):

The authors have adequately addressed reviewers comments